# Selective YAP activation in Procr cells is essential for ovarian stem/progenitor expansion and epithelium repair

**Jingqiang Wang[1,2†], Chunye Liu[2†], Lingli He[2†], Zhiyao Xie[2], Lanyue Bai[2], Wentao Yu[2], Zuoyun Wang[3], Yi Lu[2], Dong Gao[2], Junfen Fu[1]\*, Lei Zhang[2,4,5]\*, Yi Arial Zeng[2,4]\***

[1]Children's Hospital, Zhejiang University School of Medicine, National Clinical Research Center for Child Health, National Children's Regional Medical Center, Hangzhou, China; [2]State Key Laboratory of Cell Biology, CAS Center for Excellence in Molecular Cell Science, Institute of Biochemistry and Cell Biology, Chinese Academy of Sciences, University of Chinese Academy of Sciences, Shanghai, China; [3]Human Anatomy & Histoembryology, School of Basic Medical Sciences, Shanghai Medical College, Fudan University, Shanghai, China; [4]School of Life Science, Hangzhou Institute for Advanced Study, University of Chinese Academy of Sciences, Chinese Academy of Sciences, Hangzhou, China; [5]School of Life Science and Technology, ShanghaiTech University, Shanghai, China

**\*For correspondence:**
fjf68@zju.edu.cn (JF);
rayzhang@sibcb.ac.cn (LZ);
yzeng@sibcb.ac.cn (YArialZ)

[†]These authors contributed equally to this work

**Abstract** Ovarian surface epithelium (OSE) undergoes recurring ovulatory rupture and OSE stem cells rapidly generate new cells for the repair. How the stem cell activation is triggered by the rupture and promptly turns on proliferation is unclear. Our previous study has identified that Protein C Receptor (Procr) marks OSE progenitors. In this study, we observed decreased adherent junction and selective activation of YAP signaling in Procr progenitors at OSE rupture site. OSE repair is impeded upon deletion of Yap1 in these progenitors. Interestingly, Procr+ progenitors show lower expression of Vgll4, an antagonist of YAP signaling. Overexpression of Vgll4 in Procr+ cells hampers OSE repair and progenitor proliferation, indicating that selective low Vgll4 expression in Procr+ progenitors is critical for OSE repair. In addition, YAP activation promotes transcription of the OSE stemness gene *Procr*. The combination of increased cell division and Procr expression leads to expansion of Procr+ progenitors surrounding the rupture site. These results illustrate a YAP-dependent mechanism by which the stem/progenitor cells recognize the murine ovulatory rupture, and rapidly multiply their numbers, highlighting a YAP-induced stem cell expansion strategy.

## Editor's evaluation

This is a well conducted and interesting study that increases our knowledge of mechanisms governing healing after ovarian rupture. This study specifically demonstrates the importance of Procr+ progenitors that are positively regulated by YAP signalling.

## Introduction

During the adult reproductive cycles, the ovarian surface epithelium (OSE) undergoes recurring ovulatory rupture and repair (*Gaytán et al., 2005*; *Auersperg et al., 2001*). After ovulation, to maintain the physiological function and morphology of the ovary, the wound is completely closed within 12 hr to 3 days following rupture (*Burdette et al., 2006*; *Ng and Barker, 2015*; *Tan and Fleming, 2004*).

Cells surrounding the damaged sites are required to respond to the wound by turning on cell proliferation to supply sufficient cells as building blocks for regeneration (*Wang et al., 2017*). Our previous study has identified that Procr+ OSE stem/progenitor cells are the major contributor for ovulatory rupture repair. Targeted ablation of these cells hampers the repair (*Wang et al., 2019*). Interestingly, we observed that Procr+ cells expand instantly upon ovulation, reminiscent of a result of symmetric division (*Wang et al., 2019*). It remains unknown how the stem cell is triggered by the ovulation event, and what is the signal that links the rupture to the instant stem cell expansion.

The cue for this stem/progenitor cell amplification likely comes from a particular extracellular signal occurring upon ovulation. One possibility is that the follicular fluid expelled during ovulation consists of Wnts and other potential niche signals (*Ahmed et al., 2006*; *Boyer et al., 2010*; *Nilsson et al., 2001*; *Parrott et al., 2000*), which may regulate Procr+ stem/progenitor cell expansion. Another possibility is the involvement of mechanical force-induced signals, either through the change of adherent junctions or via a directed mechanism during ovulation, resulting in Procr+ stem/progenitor cell expansion.

YAP (Yes-associated protein, also known as YAP1) signaling is an evolutionarily conserved pathway and a master regulator of organ size and tissue growth during animal development (*Moya and Halder, 2019*). As a downstream effector, YAP is critical for regeneration in different organs, through triggering cell proliferation, cell survival, or expansion of stem and progenitor cell compartments (*Johnson and Halder, 2014*; *Lin et al., 2017b*; *Moya and Halder, 2016*; *Panciera et al., 2016*; *Patel et al., 2017*; *Xiao et al., 2016*; *Zhang and Del Re, 2017*). YAP is a transcriptional coactivator protein that shuttles between the cytoplasm and nucleus, and regulates the expression of target genes, such as *Ccn1* and *Birc5*, through binding with TEAD transcription factors (*Chen et al., 2001*; *Lin et al., 2017a*; *Piccolo et al., 2014*; *Totaro et al., 2018*; *Zhao et al., 2008*). Vgll4, a member of Vestigial-like proteins, serves as a transcriptional repressor of YAP through direct interactions with TEADs (*Chen et al., 2004*). Previous studies from us and others have demonstrated the important roles of Vgll4 plays during development and regeneration in various tissues (*Feng et al., 2019*; *Suo et al., 2020*; *Lin et al., 2016*; *Yu et al., 2019*). Cell–cell junctions links cells to each other in epithelial tissues, and is an upstream negative regulator of YAP (*Ramos and Camargo, 2012*; *Yang et al., 2015*). Mechanical forces regulate cell–cell adhesion stability, and cell–cell adhesion junctions may be intrinsically weak at high forces (*Pinheiro and Bellaïche, 2018*). It has been shown that disruption of adherent junctions turns on YAP nuclear activities in lung stem/progenitor cells (*Zhou et al., 2018*). However, whether YAP signaling is implicated in ovulatory rupture repair is unknown.

In this study, we investigated how OSE stem/progenitor cells are triggered by the rupture postovulation and divide subsequently. We found that, in the proximity of rupture site, decreased adherent junction is associated with increased incidence of Yap1 nuclear localization in OSE cells. Interestingly, only Procr+ OSE cells displayed a low level of Vgll4, allowing YAP signaling activation, and conditional deletion of Yap1 in Procr+ cells hampers OSE repair. We generated a new *TetO-Vgll4* mouse. Ectopic expression of Vgll4 in the stem/progenitor cells using *Procr-rtTA;TetO-Vgll4* mice blocked OSE ovulatory repair. Moreover, we found that YAP signaling activation resulted in Procr+ cells expansion at the rupture site, through the combination of inducing cell division, and directly activating Protein C Receptor (Procr) transcription. The activation of Procr is essential, as when Procr was deleted, stemness property was lost and OSE repair was hindered.

## Results

### Decreased E-cadherin expression at the rupture site and selective activation of YAP signaling in Procr+ cells

To investigate what could be the potential extracellular stimuli at the rupture site, we performed immunostaining of various adherent or tight junction components on ovarian sections. To increase rupture incidences, superovulation was induced by injection of pregnant mare serum gonadotropin (PMSG) and human chorionic gonadotropin (HCG), and the ovaries were harvested at 0.5 days after HCG injection when ovulation just occurred (*Figure 1—figure supplement 1A*). Interestingly, we found that E-cadherin, α-catenin, and ZO-1 staining is markedly decreased at the proximal region of rupture (defined as within 20 cells on one side of the rupture in section) compared to other regions, that is, rupture distal region (*Figure 1A* and *Figure 1—figure supplement 1C*) and nonrupture region

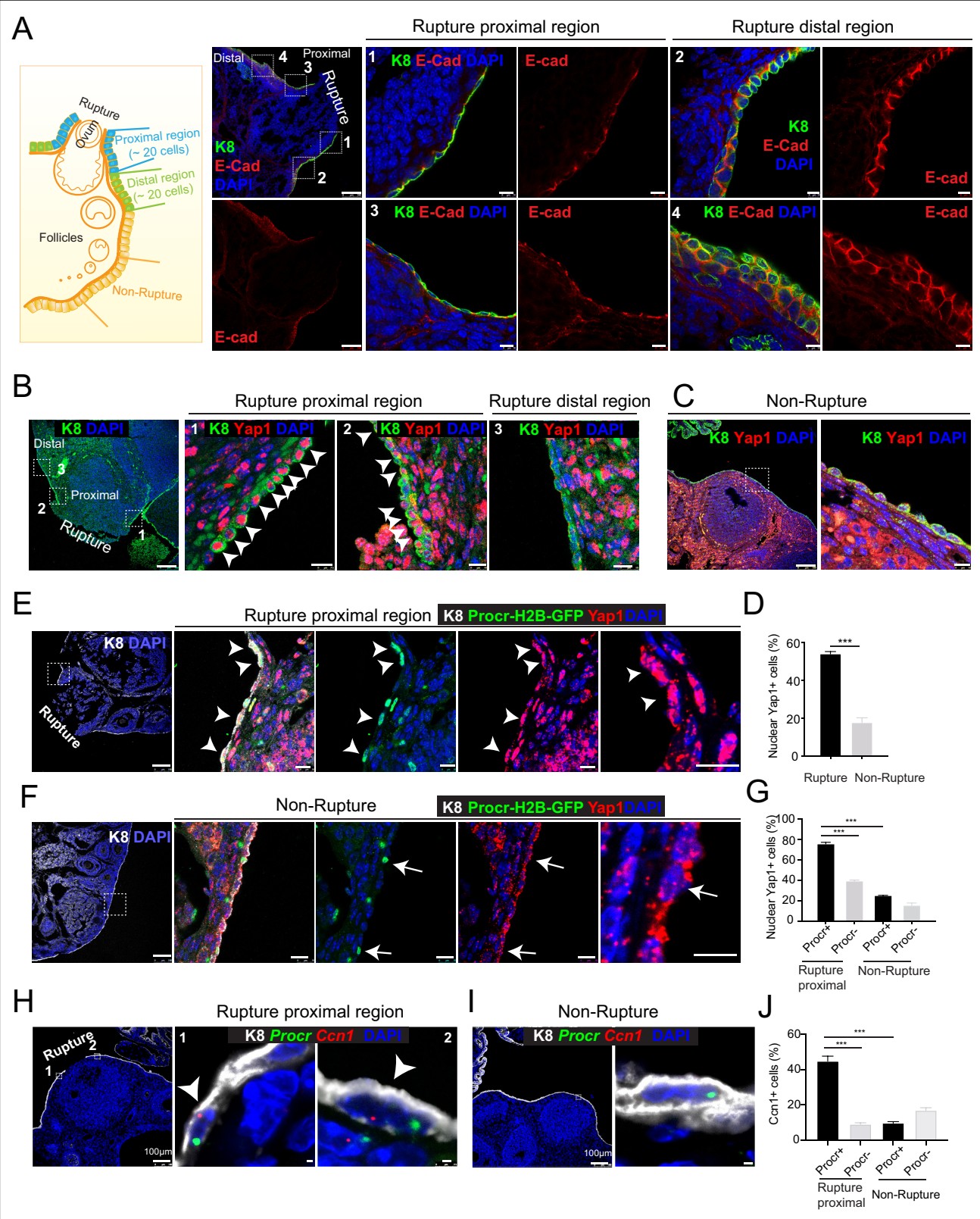

**Figure 1.** Rupture-induced YAP signaling activation is preferentially activated in Procr+ progenitors at the rupture sites. (**A**) Sections from wild-type ovaries at ovulation stage were stained with Krt8 (K8) and E-cadherin (E-cad). Confocal images showed less E-cad in the ovarian surface epithelium (OSE) of proximal regions surrounding the rupture sites (views #1, #3 in A) compared with distal regions (views #2, #4 in A). Scale bar, 100 μm for zoom out and 10 μm for zoom in. *n* = 3 mice and 15 images. (**B–D**) Sections from wild-type ovaries at ovulation stage were stained with K8 and Yap1. Confocal

*Figure 1 continued on next page*

*Figure 1 continued*

images (**B, C**) and quantification (**D**) showed Yap1 nuclear localization in the OSE was only observed in the proximal regions surrounding the rupture sites (**B, D**), but not in the distal regions (**B, D**) or the nonrupture sites (**C, D**). Scale bar, 100 μm for zoom out and 20 μm for zoom in. *n* = 3 mice and 15 images. Unpaired two-tailed *t*-test is used for comparison. ***p < 0.001. (**E–G**) *Procr-rtTA;TetO-H2B-GFP*$^{+/-}$ mice were fed with doxycycline for 3 days and harvested at ovulation stage. Confocal images of ovarian sections with K8 and Yap1 staining (E, F) and quantification (G) were showed. Nuclear Yap1 staining is preferentially detected in Procr+ (histone 2B-GFP+) cells in rupture proximal region (arrowheads in E), whereas at the nonrupture site, Yap1 staining was cytoplasmic regardless in Procr+ (arrows in F) or Procr− cells (F). Scale bar, 100 μm for zoom out and 10 μm for zoom in. *n* = 3 mice and 15 images. One-way analysis of variance (ANOVA) with Tukey test is used for comparison of multiple groups. ***p < 0.001. (**H–J**) Combination of *Procr* and *Ccn1* double fluorescent in situ with K8 antibody immunohistochemistry staining (**H–I**). Confocal images showed colocalization of *Procr* and *Ccn1* in the OSE at the rupture sites (arrowhead in H), while at nonrupture regions, both Procr+ and Procr− cells had low incidence of *Ccn1* expression (I). Quantification showed increased *Ccn1* expression in Procr+ cells at rupture sites compared with Procr− cells at rupture sites or Procr+ cells at nonrupture regions (J). Scale bar, 100 μm for zoom out and 1 μm for zoom in. *n* = 3 mice and 15 images. One-way ANOVA with Tukey test is used for comparison of multiple groups. ***p < 0.001.

The online version of this article includes the following source data and figure supplement(s) for figure 1:

**Source data 1.** Numeric data for *Figure 1D, G, J*.

**Figure supplement 1.** Decreased adherent junctions at ovarian surface epithelium (OSE) of rupture sites.

**Figure supplement 2.** Increased YAP signaling activity in Procr+ ovarian surface epithelium (OSE) cells at rupture sites.

**Figure supplement 2—source data 1.** Numeric data for *Figure 1—figure supplement 2C, D*.

(*Figure 1—figure supplement 1B,D*). As adherent junction has been implicated as a modulator of YAP signaling (*Kim et al., 2011*; *Schlegelmilch et al., 2011*; *Varelas et al., 2010*; *Yang et al., 2015*), we examined YAP activities at the rupture area by immunostaining. We observed an increased incidence of nuclear Yap1 at the proximal region of rupture compared to other regions (*Figure 1B–D*). These results suggest that compromised adherent junctions resulting from ovulatory rupture are associated with Yap1 nuclear localization in OSE cells surrounding the wound.

Our previous study has established that Procr+ progenitor cells surrounding the wound instantly proliferate upon rupture and are responsible for OSE repair (*Wang et al., 2019*). We therefore investigated whether Procr+ cells close to the rupture site are associated with YAP signaling activities. We performed Yap1 immunostaining using *Procr-rtTA;tetO-H2B-GFP* reporter, in which H2B-GFP signal marks Procr-expressing cells. Superovulation was induced in these animals by PMSG and HCG injections, and ovaries were harvested 0.5 days after HCG injection. We found that Procr+ (H2B-GFP+) cells at the rupture proximal region (referred to as rupture site from here on) have significantly higher nuclear Yap1 staining (75.9% ± 1.7%) compared to Procr− cells (39.6% ± 1.0%) (*Figure 1E, G*), or compared to Procr+ cells at the nonrupture region (*Figure 1F, G*). This was further validated by RNA double in situ hybridization with *Procr* and a YAP target gene *Ccn1*. We found that, at the rupture site, *Ccn1* is preferentially activated in Procr+ OSE cells, with 44.8% ± 2.9% of Procr+ cells being *Ccn1*+, which is markedly higher than that of Procr− cells (8.9% ± 0.8%) (*Figure 1H, J*). At the nonrupture site, both Procr+ and Procr− cells had a rather low expression of *Ccn1* expression (*Figure 1I, J*). Furthermore, we FACS-isolated Procr+ and Procr− OSE cells from the rupture sites (*Figure 1—figure supplement 2A*), validated the epithelial identity by Krt8 (K8) staining (*Figure 1—figure supplement 2B*). qPCR analysis indicated that Procr+ cells at rupture sites have a higher level of *Birc5* expression compared to Procr− cells (*Figure 1—figure supplement 2C*). Together, these results suggest that YAP signaling was specifically activated in Procr+ cells at the rupture site. Considering the role of YAP signaling in promoting cell proliferation, these results are in line with our previous observations that only Procr+, but not Procr−, cells at the rupture site displayed increased proliferation (*Wang et al., 2019*).

## Deletion of Yap1 in Procr+ cells hinders rupture repair and progenitor proliferation

To investigate whether YAP signaling is important for OSE repair, we deleted Yap1 specifically in Procr+ cells using *Procr-CreER;Yap1*$^{fl/fl}$ mice (Yap1-cKO). *Yap1*$^{fl/fl}$ mice was used as control (Ctrl). Tamoxifen (TAM) was administered in 4-week-old mice, following by superovulation at 2 days after TAM injection (*Figure 2A*). The impact on OSE repair by Yap1 deletion was analyzed by ovary wholemount imaging. At 4.5 dpi (ovulation), the two groups had similar ruptures (*Figure 2B, E*). At 6 dpi, Ctrl ovaries underwent rapid repairing (*Figure 2C, E*), and the OSE was completely recovered by 7.5

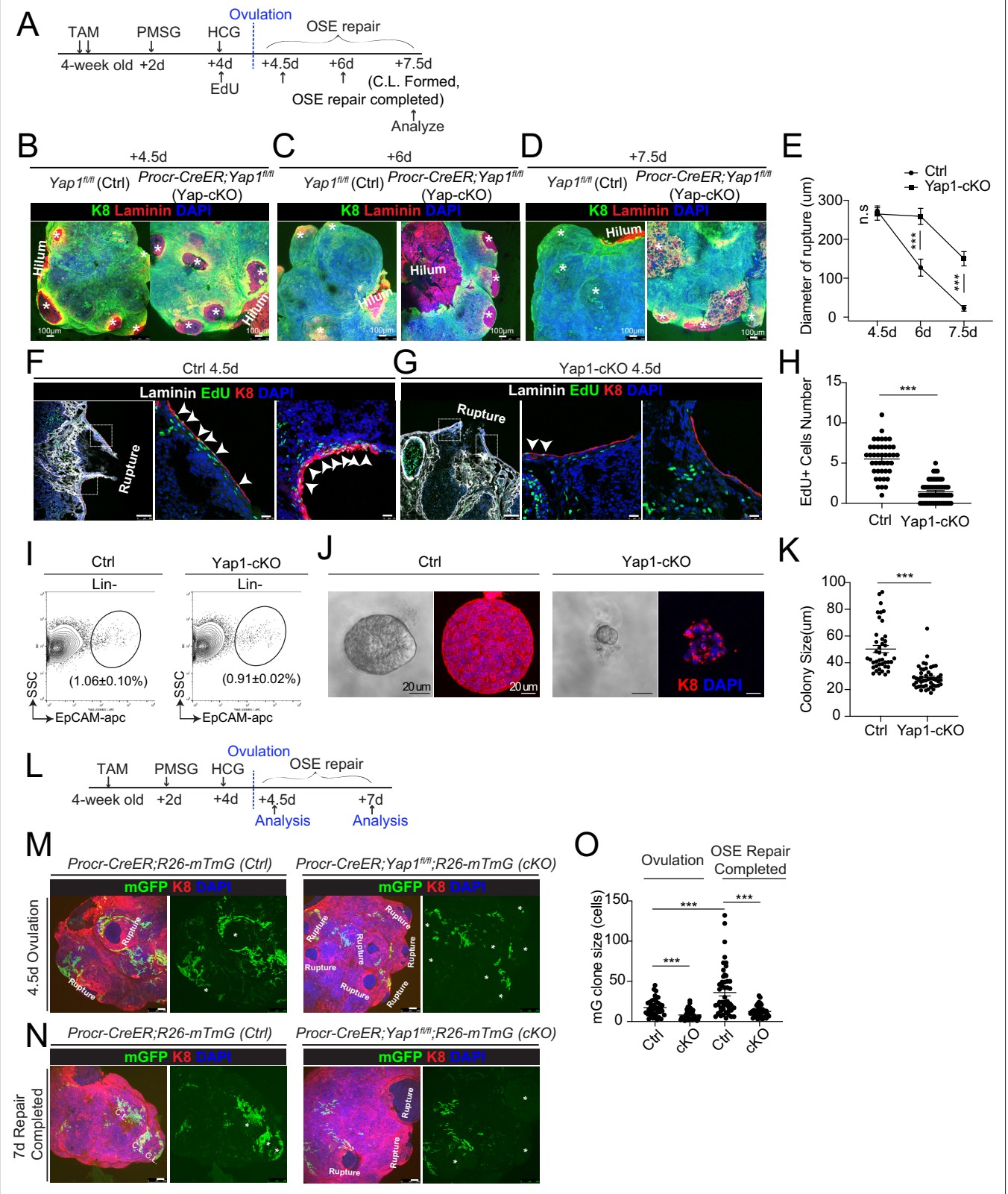

**Figure 2.** Deletion of Yap1 in Procr+ cells hinders ovarian surface epithelium (OSE) rupture repair and progenitor proliferation. (**A**) Illustration of TAM induction and superovulation strategy. (**B–E**) Yap1 was deleted in Procr+ cells using *Procr-CreER;Yap1^fl/fl^* mice (Yap1-cKO), and *Yap1^fl/fl^* mice was used as control (Ctrl). Ovary whole-mount staining with K8 and Laminin was performed (**B–D**) and the wound size in diameter was quantified (**E**). At 4.5 days (ovulation) Ctrl and Yap1-cKO ovaries had comparable wound size (* in B). At 6 days (OSE repair ongoing), the wounds in Ctrl ovary were significantly

*Figure 2 continued on next page*

*Figure 2 continued*

smaller than those in Yap1-cKO ovary ( * in **C**). At 7.5 days (repair completed), the wound was completely repaired in Ctrl, while the Yap1-cKO ovary still showed obvious wounds ( * in **D**). Scale bar, 100 μm. *n* = 3 pairs of mice. (F–H) Ctrl and Yap1-cKO mice were subjected to 12-hr 5-ethynyl-29-deoxyuridine (EdU) incorporation and were harvested at 4.5 days (ovulation stage). Representative images (**F–G**) and quantification (**H**) were showed. Out of 20 cells next to the rupture on one side, the numbers of EdU+ cells (arrowhead) in the OSE of rupture site decreased from 5.5 ± 0.3 cells in Ctrl to 1.4 ± 0.2 cells in Yap1-cKO. Scale bar, 100 μm for zoom out and 20 μm for zoom in. *n* = 3 pairs of mice. Unpaired two-tailed *t*-test is used for comparison. ***$p < 0.001$. (I-K) Total OSE cells from *Procr-CreER;Yap1$^{fl/fl}$* mice (Yap1-cKO), and *Yap$^{fl/fl}$* mice (Ctrl) were isolated by FACS at 4.5 days (ovulation stage) (**I**), followed by culture in 3D Matrigel for 7 days. Representative brightfield and confocal images of K8 staining were shown (**J**). Colony sizes in diameter were measured (**K**). Scale bar, 20 μm. Data are pooled from three independent experiments and displayed as mean ± standard error of the mean (SEM). Unpaired two-tailed *t*-test is used for comparison. ***$p < 0.001$. (L) Illustration of lineage tracing, deletion of Yap1 and superovulation strategy. (M–O) *Procr-CreER;Yap1$^{fl/fl}$;R26-mTmG* (Yap1-cKO) and *Procr-CreER;R26-mTmG* (Ctrl) mice were used. At 4.5 pi (ovulation), ovary whole-mount confocal imaging showed zones of concentrated GFP+ cells surrounding the rupture site in Ctrl, while fewer GFP+ cells were seen in Yap1-cKO ovary (**M**). At 7 pi (repair completed), ovary whole-mount confocal imaging showed large GFP+ patches located at corpus luteum (CL) in Ctrl, while rare GFP+ cells surrounding the unrepaired wound in Yap1-cKO ovary (**N**). Quantification showed significantly fewer GFP+ cells in Yap1-cKO compared with Ctrl in both ovulation stage and repair completed stage (**O**). Quantification showed an expansion of GFP+ cell numbers in Ctrl mice during the tracing and no expansion in Yap1-cKO (**O**). Scale bar, 100 μm. *n* = 3 pairs of mice. ***$p < 0.001$.

The online version of this article includes the following source data for figure 2:

**Source data 1.** Numeric data for *Figure 2E, H, K, O*.

dpi (*Figure 2D, E*). In contrast, the OSE repair in Yap1-cKO ovaries was significantly delayed at both 6 and 7.5 dpi (*Figure 2C–E*). The efficacy of Yap1 deletion and the reduced expression of the target gene *Ccn1* in OSE cells were validated by qPCR analyses (*Figure 1—figure supplement 2D*).

To analyze the proliferative capacity of Procr+ OSE cells, mice were subjected to 12 hr of 5-ethynyl-29-deoxyuridine (EdU) incorporation before harvesting the ovaries (*Figure 2A*). When analyzed at 4.5 dpi (ovulation), the number of proliferating OSE cells at rupture site (defined as 20 cells on one side from the opening) was significantly decreased from 5.5 ± 0.3 EdU+ in Ctrl to 1.4 ± 0.2 EdU+ in Yap1-cKO (*Figure 2F–H*). The impact to cell proliferation was further analyzed in vitro. Our previous study has established that Procr+, but not Procr−, OSE cells can form colonies in vitro (*Wang et al., 2019*). At 4.5 dpi, total OSE cells were isolated from both Ctrl and Yap1-cKO mice (*Figure 2I*), and placed in culture as previously described (*Wang et al., 2019*). Deletion of Yap1 in Procr+ cells drastically inhibited OSE colony formation (*Figure 2J, K*).

To visualize the contribution of Procr+ progenitors toward the repair in the presence or absence of Yap1, we performed in vivo lineage tracing. TAM was administered to 4-week-old mice to simultaneously delete Yap1 and initiate lineage tracing in Procr+ cells (*Figure 2L*). At 4.5 dpi, control (*Procr-CreER;R26-mTmG*) ovary displayed a zone of mGFP+ cells that are the progeny of Procr+ progenitors surrounding the rupture sites (*Figure 2M*). In contrast, Yap1-cKO (*Procr-CreER;Yap1$^{fl/fl}$;R26-mTmG*) ovaries have markedly fewer mGFP+ cells around the wound (*Figure 2M, O*), supporting the notion that the activity of Procr+ progenitors was hampered at the beginning of the repairing process. At 7 dpi, control ovaries had generated patches of mGFP+ cells covering the newly formed corpus luteum (*Figure 2N*). However, Yap1-cKO ovaries still had obvious openings with few mGFP+ cells (*Figure 2N, O*). Together, these results suggest that YAP signaling activation is crucial for the proliferation of Procr+ progenitor cells and the timely repair of OSE after rupture.

## An intrinsic lower level of Vgll4 in Procr+ cells is essential for their progenitor property and OSE rupture repair

Next, we investigated what could be the reason that YAP signaling is specifically activated in Procr+ cells. Vgll4 is a negative regulator of YAP by inhibiting the binding of YAP and TEAD4 (*Feng et al., 2019*). We FACS-isolated Procr+ cells and Procr− cells from the rupture sites (*Figure 3A*). qPCR analysis indicated that Procr+ cells have lower level of *Vgll4* compared to Procr− cells (*Figure 3B*). This was further validated by Vgll4 immunostaining using *Procr-rtTA;tetO-H2B-GFP* reporter, in which H2B-GFP signals mark Procr-expressing cells. Consistent with the qPCR results, Procr+ cells also exhibited lower Vgll4 protein expression compared to Procr− cells (*Figure 3C–E*).

To examine whether the reduced level of Vgll4 is significant for the selective YAP signaling activation in Procr+ cells and rupture repair, we set to overexpress Vgll4 specifically in Procr+ cells. A new *TetO-Vgll4* mouse line was generated, by inserting a tetO-Vgll4-Flag-wpre-polyA cassette behind

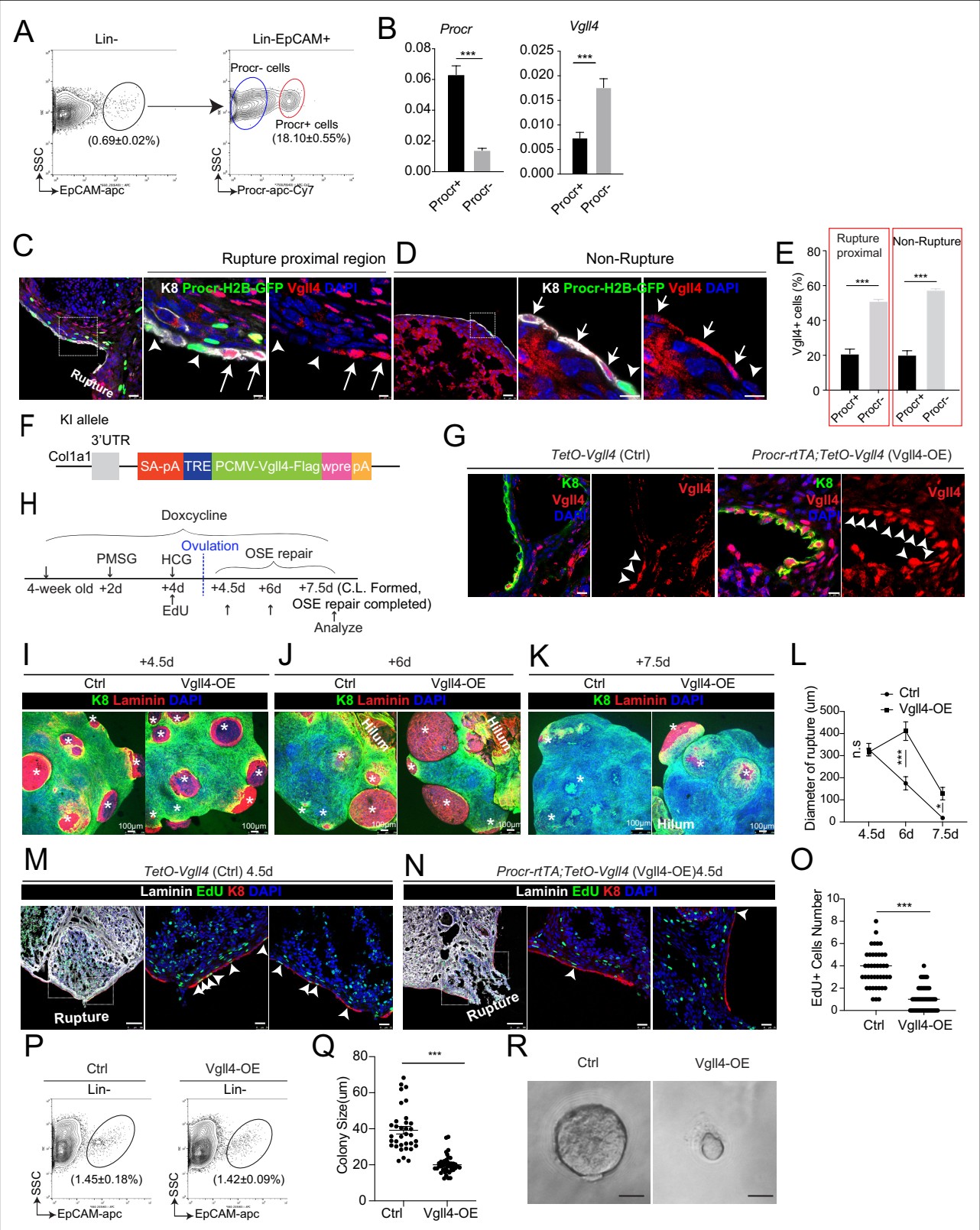

**Figure 3.** An intrinsic lower level of Vgll4 in Procr+ cells is essential for Procr+ cells' stemness and ovarian surface epithelium (OSE) rupture repair. At ovulation stage, Procr+ and Procr− OSE cells (Lin−, EpCAM+) were FACS isolated (**A**). qPCR analysis showed the lower *Vgll4* level in Procr+ cells (**B**). Data are pooled from three independent experiments and presented as mean ± standard error of the mean (SEM). ***p < 0.001. (**C–E**) *Procr-rtTA;TetO-H2B-GFP* mice were administered with pregnant mare serum gonadotropin (PMSG) and human chorionic gonadotropin (HCG) to induce superovulation,

*Figure 3 continued on next page*

 Research article

Developmental Biology

*Figure 3 continued*

and fed with doxycycline for 3 days before harvest. Ovarian sections were stained with Vgll4 and K8. Representative images showed that at both rupture proximal region (**C**) and nonrupture region (**D**), H2B-GFP− (Procr−) OSE cells have high Vgll4 expression (arrows in C, D), while H2B-GFP+ (Procr+) OSE cells have no Vgll4 expression (arrowheads in C, **D**). Scale bar, 20 μm for zoom out and 5 μm for zoom in. Quantification of the staining was shown in (**E**). $n = 3$ mice and 15 images. Unpaired two-tailed *t*-test is used for comparison. ***$p < 0.001$. Targeting strategy and validation of *TetO-Vgll4* knock-in mouse (**F, G**). A cassette of TetO-Vgll4-Flag-wpre-polyA was knocked in behind 3′UTR of *Col1a1* gene (**F**). Immunohistochemistry staining of Vgll4 in the ovaries indicated more Vgll4+ OSE cells at the rupture sites (**G**). Scale bar, 10 μm. $n = 3$ pairs of mice. Illustration of superovulation and overexpression of Vgll4 in Procr+ cells (**H**). Ovary whole-mount confocal images of K8 and Laminin showed that at 4.5 days (ovulation), Ctrl (*TetO-Vgll4*) and Vgll4-OE (*Procr-rtTA;TetO-Vgll4*) ovaries have similar wound sizes (* in I). At 6 days (repair ongoing), the wound sizes in Ctrl mice were smaller than those in Vgll4-OE (* in J). At 7.5 days (repair completed), Ctrl ovary had completely repaired, while Vgll4-OE mice had obvious opening (* in K). Scale bar, 100 μm. The sizes of the wound in diameter were quantified (**L**). $n = 3$ pairs of mice. The mice were harvested at 4.5 days (ovulation) after 12-hr 5-ethynyl-29-deoxyuridine (EdU) incorporation. Representative images (**M, N**) and quantification (**O**) showed the number of EdU+ cells in the OSE surrounding the rupture site (arrowheads in M) decreased from 3.7 ± 0.3 in Ctrl to 1.0 ± 0.2 in Vgll4-OE (arrowheads in N). Scale bar, 100 μm for zoom out and 20 μm for zoom in. $n = 3$ pairs of mice. Unpaired two-tailed *t*-test is used for comparison. ***$p < 0.001$. Total OSE cells were isolated by FACS from Ctrl and Vgll4-OE at 4.5 days (ovulation) (**P**), and cultured in 3D Matrigel. At day 7 in culture, colony sizes were measured in diameter (**Q**), and representative images were shown (**R**) out of 15 images in each group. Scale bar, 20 μm. Data are pooled from three independent experiments and displayed as mean ± SEM. Unpaired two-tailed *t*-test is used for comparison. ***$p < 0.001$.

The online version of this article includes the following source data and figure supplement(s) for figure 3:

**Source data 1.** Numeric data for *Figure 3B, E, L, O, Q*.

**Figure supplement 1.** Construction of *TetO-Vgll4* mouse model.

**Figure supplement 1—source data 1.** Numeric data for *Figure 3—figure supplement 1E*.

**Figure supplement 1—source data 2.** Whole blot image for *Figure 3—figure supplement 1B–D*.

the 3′UTR of the *Col1a1* gene (*Figure 3F* and *Figure 3—figure supplement 1A–C*). Subsequently, *Procr-rtTA;TetO-Vgll4* (Vgll4-OE) mice were generated by genetic crosses with *TetO-Vgll4* as control (Ctrl). The efficacy of overexpression was validated by western blotting and qPCR, showing increased expression of Vgll4 and decreased expression of *Ccn1* in Vgll4-OE cells (*Figure 3—figure supplement 1D, E*). Furthermore, immunostaining confirmed the increased number of Vgll4 high cells in the OSE layer of Vgll4-OE mice (*Figure 3G*). For this experiment, superovulation was performed to 4-week-old mice and doxycycline hyclate (DOX) was fed throughout the process (*Figure 3H*). The impact of Vgll4 overexpression was analyzed throughout the repairing process, at 4.5 dpi (ovulation), 6 dpi (OSE repair ongoing), and 7.5 dpi (OSE repair completed) by ovary whole-mount imaging. We found that the rupture in Ctrl and Vgll4-OE ovaries are comparable at 4.5 dpi (*Figure 3I, L*). At 6 dpi, while Ctrl ovaries had sights of wound closure, Vgll4-OE ovaries still showed larger areas of rupture (*Figure 3J, L*). At 7.5 dpi, Ctrl ovaries displayed complete OSE, whereas the repair in Vgll4-OE ovaries was obviously delayed (*Figure 3K, L*).

Next, we examined whether overexpression of Vgll4 affects progenitor proliferation. At 4.5 dpi (ovulation), the number of proliferating OSE cells at rupture site was significantly decreased from 3.7 ± 0.3 in Ctrl to 1.0 ± 0.2 in Vgll4-OE (*Figure 3M–O*). At 4.5 dpi, total OSE cells were isolated and cultured in vitro for 7 days (*Figure 3P*). Consistently, overexpression of Vgll4 inhibits cell proliferation and colony formation (*Figure 3Q, R*). Together, these results suggest that overexpression of Vgll4 in Procr+ cell impaired Procr+ cell proliferation and ovulatory rupture repair.

## YAP signaling promotes Procr+ cells expansion at rupture site

We have previously found that Procr+ progenitor cells expand instantly at the periphery of the rupture site upon ovulation (*Wang et al., 2019*). To investigate whether YAP signaling activation is linked to the expansion of Procr+ progenitor cells, TAM was administered to 4-week-old *Procr-CreER;Yap1^{fl/fl}* (Yap1-cKO) and *Yap1^{fl/fl}* (Ctrl) mice for two times, followed by superovulation. At 4.5 dpi (ovulation), FACS analysis showed a dramatic decrease of Procr+ progenitor population when Y*ap1* was deleted (*Figure 4A–C*).

To better visualize the change of Procr+ progenitor cells under the influence of YAP signaling, we generated *Procr-rtTA;TetO-H2B-GFP^{+/−};TetO-Vgll4^{+/−}* mice (Vgll4-OE). Superovulation was performed to 4-week-old mice and DOX was fed throughout the experiments to maintain the expression of H2B-GFP in Procr+ cells (*Figure 4D*). When analyzed at 4.5 dpi (ovulation), at the wound edge (defined as 20 cells on one side from the opening) of control ovary (*Procr-rtTA;TetO-H2B-GFP^{+/−}*), there were about

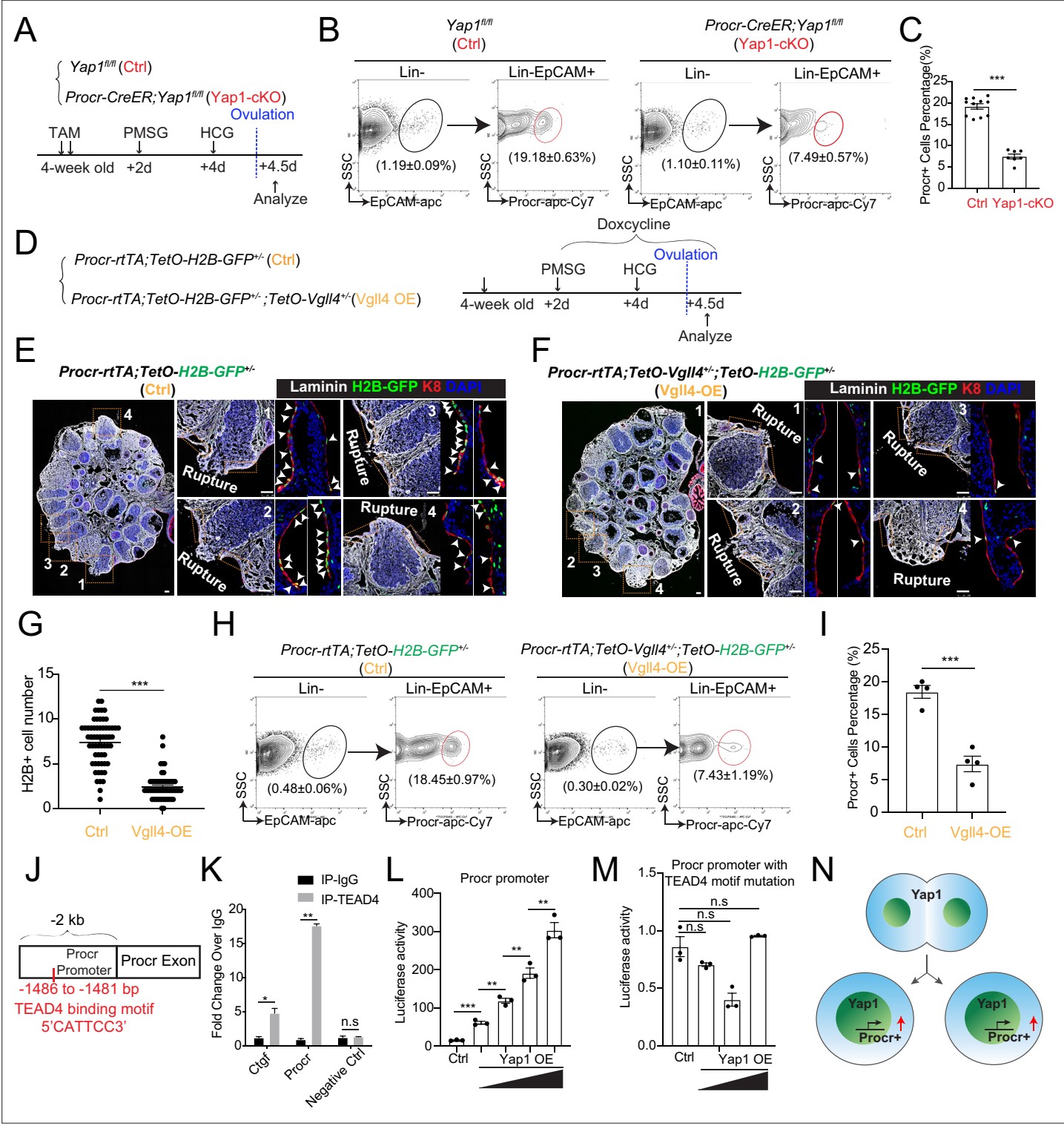

**Figure 4.** YAP signaling promotes Procr+ cells expansion at rupture sites through a combination of promoting cell division and enhancing Procr expression. Illustration of superovulation and analysis strategy as indicated using *Yap1*[fl/fl] (Ctrl) and *Procr-CreER;Yap1*[fl/fl] (Yap1-cKO) mice (**A**). At ovulation stage, the percentage of Procr+ ovarian surface epithelium (OSE) cells in Ctrl and Yap1-cKO were FACS analyzed (**B**) and quantified (**C**). *n* = at least three mice in each group and displayed as mean ± standard error of the mean (SEM). Unpaired two-tailed *t*-test is used for comparison. ***p < 0.001. Illustration of superovulation and analysis strategy as indicated using *Procr-rtTA;TetO-H2B-GFP*[+/−] (Ctrl) and *Procr-rtTA;TetO-H2B-GFP*[+/−];*TetO-Vgll4*[+/−] (Vgll4-OE) mice (**D**). At ovulation stage, ovary section imaging showed that at the rupture sites, the number of H2B-GFP+ (Procr+) cells in Ctrl (arrowheads in E) is higher than those in Vgll4-OE (arrowheads in F). Scale bar, 100 µm. Quantification was shown in (**G**). *n* = 3 pairs of mice and

*Figure 4 continued on next page*

*Figure 4 continued*

15 images in each group. Unpaired two-tailed *t*-test is used for comparison. \*\*\*p < 0.001. The percentage of Procr+ OSE cells were analyzed by FACS at ovulation stage (**H**). The percentage of Procr+ cells in Ctrl are higher than that in Vgll4-OE (**H, I**). *n* = at least 3 mice and displayed as mean ± SEM. Unpaired two-tailed *t*-test is used for comparison. \*\*\*p < 0.001. Illustration of Tead4 motif in Procr promoter region (**J**). TEAD4 chromatin immunoprecipitation-qPCR (ChIP-qPCR) analysis using cultured primary OSE cells showed the enrichment of Procr promoter, and Ctgf promoter was used as positive control (**K**). *n* = 2 biological repeats. Unpaired two-tailed *t*-test is used for comparison. \*\*p < 0.01, \*p < 0.05, n.s., not significant. Analysis of luciferase reporter activity driven by WT (**L**) and Tead4 motif (−1486 to −1481 bp) deleted Procr promoter (**M**) in HEK293T cells transfected with increased amount of Yap1 overexpression plasmids. Data are pooled from three independent experiments and displayed as mean ± SEM. Unpaired two-tailed *t*-test is used for comparison. \*\*\*p < 0.001, \*\*p < 0.01, n.s., not significant. (**N**) A proposed model of which YAP signaling promotes Procr+ cells expansion at rupture site through a combination of promoting cell division and enhancing Procr expression.

The online version of this article includes the following source data and figure supplement(s) for figure 4:

**Source data 1.** Numeric data for *Figure 4C, G, I, and K–M*.

**Figure supplement 1.** YAP promotes Procr+ cells expansion.

**Figure supplement 1—source data 1.** Numeric data for *Figure 4—figure supplement 1C*.

**Figure supplement 2.** YAP induces Procr expression.

**Figure supplement 2—source data 1.** Numeric data for *Figure 4—figure supplement 2C, E, F*.

**Figure supplement 2—source data 2.** Whole blot image for *Figure 4—figure supplement 2K*.

7.4 ± 0.3 H2B-GFP+ cells expressing the peak level of GFP (*Figure 4E, G*). In contrast, in Vgll4-OE ovary (*Procr-rtTA;TetO-Vgll4;TetO-H2B-GFP⁺/⁻*), only 2.4 ± 0.2 H2B-GFP+ cells were observed at the wound edge (*Figure 4F, G*). FACS analysis also showed that the percentage of Procr+ progenitor population decreased significantly from 18.5% ± 1.0% in Ctrl to 7.4% ± 1.2% in Vgll4-OE at ovulation stage (*Figure 4H, I*).

The proliferative activity of Procr+ cells was further evaluated in vitro. We isolated OSE cells from control (*Procr-rtTA;TetO-H2B-GFP⁺/⁻*) and Vgll4-OE (*Procr-rtTA;TetO-Vgll4;TetO-H2B-GFP⁺/⁻*) mice and placed in culture, followed by live imaging to document the division of H2B-GFP+ (Procr+) cells (*Figure 4—figure supplement 1A, B*). In control cells, we observed frequent division of Procr+ cells, and in most cases, it was one Procr+ cell dividing into two Procr+ cells (*Figure 4—figure supplement 1A, C*). But in Vgll4-OE, we could hardly observe cell division (*Figure 4—figure supplement 1B, C*). Together, these results suggest that inhibition of YAP signaling, by either Yap1-deletion or Vgll4-OE, impairs the expansion of Procr+ progenitors upon rupture.

## YAP signaling enhances Procr expression

It is unclear how YAP maintains Procr expression during or after cell division. Thus, we investigated the association of YAP activation and Procr expression. OSE cells were isolated from *Procr-rtTA;TetO-H2B-GFP⁺/⁻* mice, and cultured on glass (YAP activation) or soft condition (0.48 kPa, YAP inactivation) (*Figure 4—figure supplement 2A, B*). DOX was added 2 days before harvest. Consistent with the notion, we found that, in soft condition, Yap1 was mostly cytoplasmic and most OSE cells are H2B-GFP− (*Figure 4—figure supplement 2A*). In contrast, most OSE cells are H2B-GFP+ in stiff condition and Yap1 was found in the nucleus (*Figure 4—figure supplement 2B*). These observations suggest that YAP activation might induce Procr expression. We verified by qPCR that *Procr* expression is upregulated in stiff conditions (*Figure 4—figure supplement 2C*). Our results support the notion that YAP activation induces Procr expression.

To further investigate whether Yap1 regulates Procr expression, we knocked down Yap1 by shRNA in OSE culture and found that this inhibits Procr expression (*Figure 4—figure supplement 2D, E*). Furthermore, blocking YAP activation by Verteporfin (VP) or Vgll4 overexpression also resulted in lower Procr expression (*Figure 4—figure supplement 2F, G*). These results suggest that inhibiting YAP signaling suppresses Procr expression.

To investigate whether YAP/TEAD4 directly regulate Procr expression, we analyzed the promoter of *Procr*. A Tead4-binding motif (5′-CATTCC-3′) was found at the proximal promoter of *Procr* (−1486 to −1481 bp) (*Figure 4J*). Chromatin immunoprecipitation-qPCR (ChIP-qPCR) showed that Tead4 could directly bind to the *Procr* promoter (*Figure 4K*). Therefore, we examined whether this Tead4-binding motif is responsible for induction of Procr expression by Yap1. While Yap1 induced the wild-type promoter luciferase in a dose-dependent manner (*Figure 4L* and *Figure 4—figure supplement 2H,*

*I*), it could not activate the mutant reporter with the deletion of the Tead4-binding motif (*Figure 4M* and *Figure 4—figure supplement 2H, I*). These results suggest that Yap1 directly promotes Procr expression. Together, our data support a model that YAP signaling promotes expansion of Procr+ cells at rupture site through a combination of increased cell division and Procr expression (*Figure 4N*).

## Procr is essential for the progenitor property

The upregulation of Procr expression coupled with YAP-induced cell division implies that the expression of Procr may be important for keeping the stem cell property in OSE. To assess the significance of Procr, we utilized a *Procr-flox* allele (Liu and Zeng, unpublished) and specifically deleted Procr in the progenitor using *Procr$^{CreER/fl}$* (Procr-cKO) mice. TAM was administered in 4-week-old mice for two times, followed by superovulation at 2 days after TAM injection (*Figure 5A*), and the phenotype was analyzed by ovary whole-mount imaging. Ctrl and Procr-cKO ovaries formed comparable ruptures at 4.5 dpi (ovulation) (*Figure 5B, E*). At 6 dpi (OSE repair ongoing), control ovaries showed smaller openings compared to Procr-cKO (*Figure 5C, E*). At 7.5 dpi (OSE repair completed), Ctrl ovaries were covered by complete OSE, whereas Procr-cKO ovaries still had regions with unrepaired OSE (*Figure 5D, E*). Furthermore, at 4.5 dpi (ovulation), the ovaries were harvested after 12 hr of EdU incorporation. The number of proliferated OSE cells at rupture site decreased from 4.7 ± 0.4 cells in control (*Procr$^{fl/+}$*) to 1.6 ± 0.2 cells in Procr-cKO (*Figure 5F–H*). After deletion of Procr in vivo, total OSE cells were isolated and cultured in vitro for 7 days (*Figure 5I*). We found that deletion of Procr inhibits the proliferation of progenitor cells, resulting in reduced colony sizes (*Figure 5J–L*). Overall, these data suggest that Procr is essential for progenitor property upon rupture.

## Discussion

In this study, we addressed the molecular mechanism which links the ovulatory rupture to OSE stem/progenitor cells activation, thus promptly turning on proliferation and repairing the wound. Our findings support the following model. Procr+ OSE progenitors have intrinsically lower levels of Vgll4. Upon ovulatory rupture, the decreased adherent junction at the proximity of the rupture site promotes Yap1 nuclear localization. These intrinsic and extrinsic factors together lead to YAP signaling activation in Procr+ progenitors around the wound, which sequentially stimulates the proliferation of the progenitors. Importantly, YAP activation directly upregulates Procr expression in the dividing cells, resulting in the expansion of Procr+ progenitors around the wound (*Figure 5M*). Blocking YAP signaling in the progenitors by Yap1-cKO or Vgll4-OE impairs the progenitors' activities and hinders OSE repair. Furthermore, Procr function is essential for these progenitors. When Procr was deleted, stem cell property was lost hindering OSE repair.

While we uncovered the significance of selective activation of YAP in OSE progenitors, it is still unclear how ovarian rupture is sensed and how YAP signaling is induced by injury. We observed reduced cell-adhesion protein E-cadherin is lowered at the site of the ovarian rupture, and it is previously known that E-cadherin signals through the Hippo pathway to block YAP (*Yang et al., 2015*). The current missing link is why and how E-cadherin would be reduced at the ovarian rupture site. We speculate that during the late stage of follicle development, the pre-ovulatory follicle forms a protrusion toward OSE. Subsequently, ovulation generates a rupture on OSE. These contiguous events likely induce the thinning of OSE surrounding the pre-ovulatory follicles and at the proximity of the rupture site, resulting in the reduced adherent junction proteins. Yet, there are other potentially more direct possibilities, i.e. mechanical stretching induces YAP (*Halder et al., 2012*). First, the pre-ovulatory follicle protrusion or the release of oocytes induces a mechanical force on the OSE surrounding the wound, activating YAP signaling. Second, at the rupture site, epithelial cells no longer become packed together because the epithelium has been denuded, therefore potentially cells flanking the rupture site would become 'stretched', consequently actives YAP.

YAP signaling promotes Procr+ cell expansion at the rupture site through a combination of increased cell division and Procr expression. In the current study, YAP is particularly activated in Procr+ progenitor cells at the rupture site. We observed that at the rupture sites, Vgll4 is highly expressed in Procr– cells, preventing YAP pathway activation in those cells around the rupture sites. Our findings demonstrate that the reduced levels of Vgll4 in Procr+ progenitors likely contribute to the selective

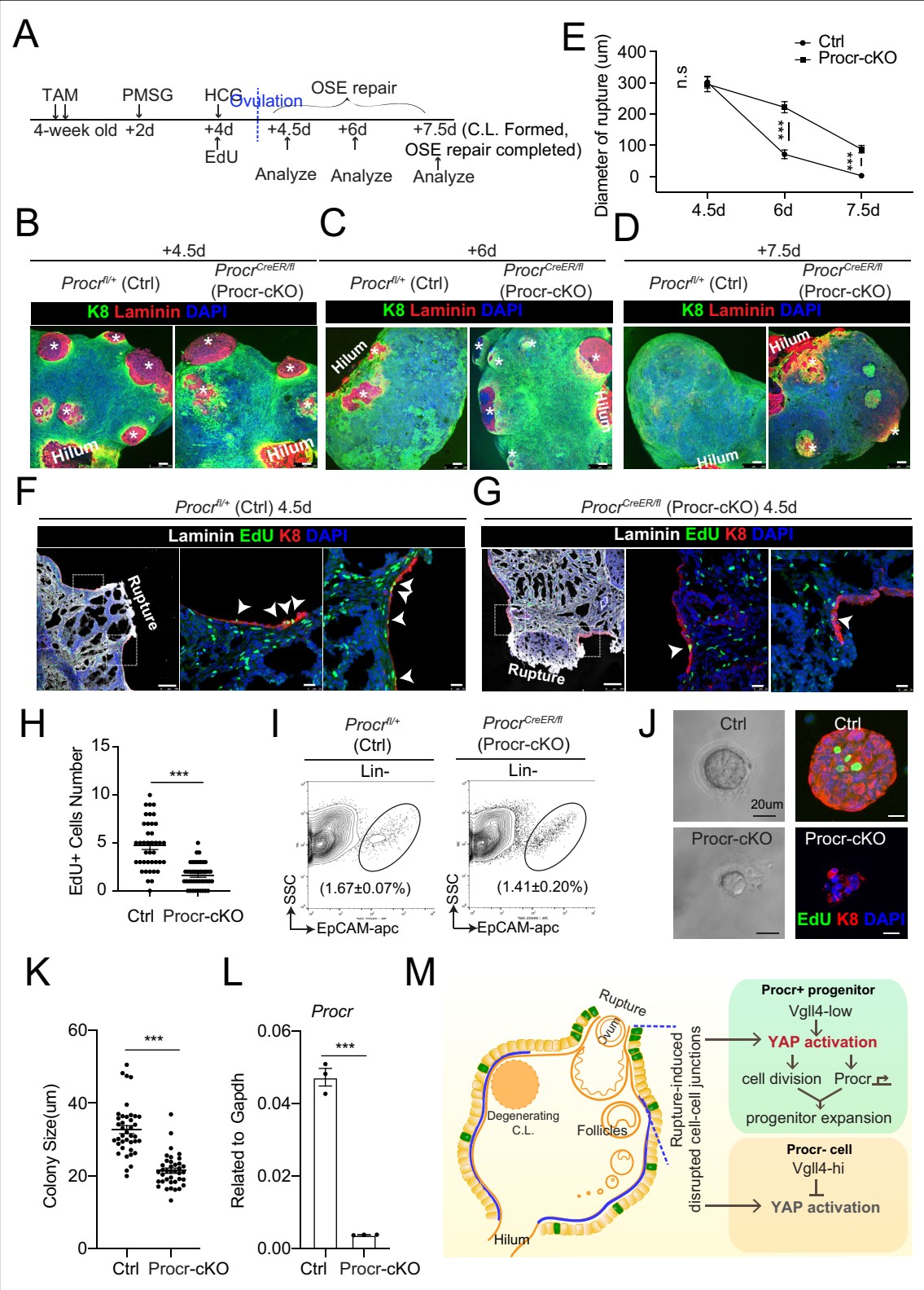

**Figure 5.** Procr is essential for the progenitor property. (A–E) Illustration of superovulation and deletion of *Procr* in Procr+ cells using *Procr*CreER/fl mice (Procr-cKO), and *Procr*fl/+ mice (Ctrl) (**A**). Ovary whole-mount confocal imaging of K8 and Laminin showed that at 4.5 days (ovulation), Ctrl and Procr-cKO have similar wound sizes (* in B). At 6 days (ovarian surface epithelium [OSE] repair ongoing), the wound sizes in Ctrl mice were smaller than those in Procr-cKO (* in C). At 7.5 days (repair completed), Ctrl ovary had completely repaired, while Procr-cKO remained obvious opening (* in D). Scale bar, 100 μm. Quantification of the wound size in diameter was shown in (**E**). *n* = 3 pairs of mice. Unpaired two-tailed *t*-test is used for comparison. ***p < 0.001.

*Figure 5 continued on next page*

*Figure 5 continued*

n.s., not significant. (F–H) Post 12-hr 5-ethynyl-29-deoxyuridine (EdU) incorporation, the mice were harvested at 4.5 days (ovulation) (**A**). Representative images showed the number of EdU+ cells (arrowhead) in the OSE surrounding the rupture site decreased from 4.7 ± 0.4 in Ctrl (arrowheads in F) to 1.6 ± 0.2 in Procr-cKO (arrowheads in G). Scale bar, 100 μm for zoom out and 20 μm for zoom in. Quantification of was shown in (**H**). $n$ = 3 pairs of mice. Unpaired two-tailed $t$-test is used for comparison. \*\*\*p<0.001. Total OSE cells from Ctrl andProcr-cKO were isolated by FACS (**I**), followed by culture in 3D Matrigel. At culture day 7, representative brightfield and confocal images with K8 staining showed that OSE cells with Procr-cKO form markedly smaller colonies compared to Ctrl (**J**). Colony sizes were quantified in (**K**). qPCR analysis validated the deletion efficiency of *Procr* in OSE cells of Procr-cKO (**L**). Data are pooled from three independent experiments and displayed as mean ± standard error of the mean (SEM). Unpaired two-tailed $t$-test is used for comparison. \*\*\*p < 0.001. Scale bar, 20 μm. (M) A proposed model of YAP activation in Procr+ cells promoting OSE progenitor cell expansion. Procr+ OSE progenitors have intrinsically lower level of Vgll4 compared to Procr− OSE cells. At ovulation, cell–cell junctions at rupture site were disrupted, which induces the possibility of YAP activation in all OSE cells surrounding the rupture. However, the lower expression of Vgll4 in Procr+ cells allowed YAP activation in the progenitor cells at this area. YAP activation in Procr+ cells promoted cell division, and importantly, it directly upregulates Procr expression in the dividing cells, resulting in expansion of Procr+ progenitors around the wound.

The online version of this article includes the following source data for figure 5:

**Source data 1.** Numeric data for *Figure 5E, H, K, L*.

activation of YAP signaling in these cells. Further study should investigate what mechanism determines the lower expression of Vgll4 in Procr+ progenitor cells.

In the current study, we generated a new *TetO-Vgll4* mouse model that enables the overexpression of Vgll4 in a specific cell type. The overexpression of Vgll4 in the progenitor of OSE has been validated using *Procr-rtTA*. The advantages brought by our *TetO-Vgll4* reporter will be of broad value in studies of Hippo-Yap signaling across all tissues.

Procr expression is initially found on the surface of vascular cells exerting an anticoagulation role, by binding and activating protein C (PC) in the extracellular compartment (*Fukudome et al., 1998*). More recently, studies from us and others have identified Procr as a stem cell surface marker in multiple tissues (*Wang et al., 2015*; *Wang et al., 2020*; *Wang et al., 2019*; *Yu et al., 2016*), but less is known regarding the function of Procr in stem/progenitor cells. In the current study, we demonstrate that, Procr is essential for the proliferation of Procr+ progenitor cells and OSE repair upon rupture. Our previous report indicated that PROCR concomitantly activates multiple pathways including ERK, PI3K–Akt–mTOR, and RhoA–Rock–P38 signaling in breast cancer cells (*Wang et al., 2018*). We speculate that similar intracellular pathways might be involved in the Procr+ OSE cells. Procr is regarded as a Wnt target gene from an in vitro screen in mammary stem cell culture (*Wang et al., 2015*). In this study, we identify YAP as a novel upstream regulator of Procr. ChIP-qPCR and promoter luciferase experiments demonstrate that *Procr* transcription can be directly upregulated by YAP activation.

The phenomena of YAP promoting stem/progenitor cell expansion have been reported in various tissues (*Beverdam et al., 2013*; *Camargo et al., 2007*; *Cao et al., 2008*; *Ramos and Camargo, 2012*; *Schlegelmilch et al., 2011*; *Zhou et al., 2018*). Yet, in this process, less is known about how YAP maintains stem cell properties. To the best of our knowledge, this is the first report illustrating a mechanism through which YAP promotes cell proliferation, and simultaneously upregulates the expression of an essential stemness gene to maintain cell fate, leading to a rapid expansion of stem cell numbers around the wound. In summary, our study provides new evidence and molecular insights into how ovulatory rupture triggers the activation of OSE stem cells, resulting inpromptly expanding their numbers for repair. This may have a broad implication to understand the action of tissue stem cells during would healing in other tissue.

## Materials and methods
### Lead contact and materials availability

Further information and requests for reagents should be directed to and will be fulfilled by the Lead Contact, Yi Arial Zeng (yzeng@sibcb.ac.cn). All unique/stable reagents generated in this study are available from the Lead Contact with a completed Materials Transfer Agreement.

### Experiment animals

*TetO-H2B-GFP*^+/− (Stock: 005104), *R26-mTmG*^+/− (Stock: 007576) from Jackson Laboratories, *Procr*^CreER (*Wang et al., 2015*), *Procr*^rtTA (*Wang et al., 2019*), *YAP*^fl/+ (*Feng et al., 2019*), *Procr*^fl/+(Liu and Zeng,

unpublished), *TetO-Vgll4* were used in this study. The *TetO-Vgll4* mouse line was generated by knocking in a cassette of TetO-Vgll4-Flag-wpre-polyA behind 3′UTR of *Col1a1* gene (**Figure 3—figure supplement 1**). All mice were housed in the SIBCB animal facility under IVC standard with a 12 hr light/dark cycle at room temperature. Both ovaries were used per mice and the number of mice per experiment was shown in figure legends. For targeted knockout in vivo, 4–5 weeks mice were administered with TAM diluted in sunflower oilby intraperioneal (IP) injection at a concentration of 2 mg per 25 g body weight for two or three times (on every second day). For superovulation experiments, 4- to 5-week-old mice were injected with 10 IU of PMSG by IP, followed by IP injection of 10 IU of HCG about 48 hr later. For DOX feeding, DOX was dissolved in drink water at a concentration of 1 mg/ml. Experimental procedures were approved by the Animal Care and Use Committee of Shanghai Institute of Biochemistry and Cell Biology, Chinese Academy of Sciences, with a project license number of IBCB0065.

## OSE cells isolation and flow cytometry

Ovaries from superovulated or 4- to 12-week-old female mice were isolated, and the oviduct and bursa were carefully cleared out under dissect microscope. The ovaries were minced into pieces as small as possible, and then placed in 10 ml digest buffer (RPMI 1640 with 5% fetal bovine serum (FBS), 1% penicillin–streptomycin, 25 mM 4-(2-hydroxyethyl)-1-piperazineethanesulfonic acid (HEPES), and 300 U/ml collagenase IV). After digestion at 37°C, 100 rpm for about 1 hr, ovarian cells were obtained after centrifugation at 1000 rpm for 5 min. The red blood cells were lysed with buffer at room temperature for 5 min, and then single cells were obtained with 0.25% trypsin treatment at 37°C for 5 min, followed by 0.1 mg/ml DNaseI incubation at 37°C for 5 min with gently pipetting before filtering through 70 µm cell strainers. The single cells were incubated on the ice and in dark with the following antibodies at a dilution of 1:200: FITC conjugated, PE conjugated, or biotinylated CD31, CD45, EpCAM-APC, Procr-PE, Procr-Biotin, Streptavidin-APC-Cy7, and Streptavidin-V450. All analysis and sorting were performed using a FACSJazz (Becton Dickinson). The purity of sorted population was routinely checked and ensured to be >95%.

## OSE cells 3D culture assay

FACS sorted OSE cells were resuspended with 60 µl 100% growth factor-reduced Matrigel and placed around the rim of a well of a 24-well plate, and allowed to solidify for at least 15 min at 37°C in a 5% $CO_2$ incubator before adding 0.5–1 ml culture medium. Colonies were grown for 7–9 days and the medium was changed every other days. The culture medium was prepared by adding 5% FBS, 4 mM L-glutamine, 1 mM sodium pyruvate, 10 ng/ml epidermal growth factor, 500 ng/ml hydrocortisone, 5 mg/ml insulin, 5 mg/ml transferrin, 5 ng/ml sodium selenite, 0.1 mM Minimum Essential Medium (MEM) nonessential amino acids, $10^{-4}$ M 2-mercaptoethanol into Dulbecco's modified essential medium (DMEM)/F12. The organoid images were captured by Zeiss inverted microscope at days 7–9.

## Immunohistochemistry

For section staining, ovarian tissues were fixed in 4% PFA at room temperature for 15 min, following by washed with phosphate-buffered saline (PBS) for three times, dehydrated in 30% sucrose at 4°C overnight and embedded with Optimum Cutting Temperature. 16–18 µm tissue sections were incubated in 0.1% or 0.5% Triton X-100 diluted with PBS (PBST) for 20 min and then 1 hr blocking using 10% FBS in PBST. Then sections were incubated with primary antibodies diluted in blocking buffer at 4°C overnight, followed by washes for three times (20 min per time). After wash, the sections were further incubated with secondary antibodies and 4′,6-diamidino-2-phenylindole (DAPI) diluted in blocking buffer for 2 hr at room temperaturein dark, followed by washes for three times (20 min per time) and mounted with mounting medium.

For staining of cultured colonies, colonies were released from Matrigel by incubating with dispase for 20–30 min. Then the colonies were fixed in 4% PFA on ice for 10 min, followed by cytospin (Thermo Fisher) into slides and staining protocol described above.

For whole mouse ovary immunohistochemistry, mouse ovaries that cleared without bursa and oviduct were fixed with fresh 4% PFA at room temperature for 15 min in 4 ml Eppendorf tubes, followed by washing with PBST for three times (20 min per time). The staining of whole ovaries was then transferred into the 2 ml Falcon tubes using a dropper carefully. Ovaries were blocked for 1 hr

using 10% FBS in PBST. Then, the ovaries were incubated with primary antibodies diluted in blocking buffer at 4°C for 48 hr on a transference shaker with 10rpm, followed by washing for three times (20min per time) at room temperature. After washing, the ovaries were incubated with secondary antibodies diluted in blocking buffer for 24 hr at 4°C in dark, and counterstained with DAPI on a transference shaker with 10rpm, followed by washing for three times (20min per time) at room temperature. The ovaries could be stored in PBST at 4°C for at least 2 weeks.

For Yap1 staining in vivo, tyramide signal amplification assay (TSA staining) with Yap1 antibody from CST (Cat# 14074) was used. Briefly, paraffin sections were rehydrated in histoclear and gradual ethanol (100%, 100%, 95%, 85%, 75%, 50%, and 30%) and the TSA staining was performed using the Opal 4-Color Automation IHC Kit (PerkinElmer) following the manufacturer's instructions. After TSA staining for Yap1, staining for GFP and Krt8 was performed following protocol described above.

Tissue sections and organoids fluorescent images were captured using Leica DM6000 TCS/SP8 laser confocal scanning microscope with a ×20/0.75 or ×40/0.75 or ×63/0.75 IMM objective with 1–3 μm z-step. Confocal images were processed with maximum intensity projections.

Whole mouse ovarian fluorescent images were captured with inverted Leica TCS SP8 WLL at a ×10/0.75 objective, z-stack was ~50–80 layers with 6–7 μm per layer, and the area was about 1.5 mm × 1.5 mm, which was about 1/6–1/4 of the ovary surface.

## Western blotting

Digested cells were lysed in sodium dodecyl sulfate–polyacrylamide gel electrophoresis (SDS–PAGE) loading buffer and boiled for 10 min. Proteins were separated by SDS–PAGE and transferred to nitrocellulose membrane (GE Company). Bolts were blocked with 3% Bovine Serum Albumin (BSA) in Tris-Buffered Saline with 0.5% Tween 20 (TBST) (50 mM Tris–HCl, 150 mM NaCl, 0.05% Tween-20, pH 7.5) for 1 hr and incubated with primary antibodies at 4°C overnight, followed by incubated with secondary IgG-HRP antibodies for 2 hr at room temperature. Protein bands were visualized with chemiluminescent reagent and exposed to Mini Chemiluminescent Imager.

## RNA in situ

In situ hybridization was performed using the RNA scope kit (Advanced Cell Diagnostics) following the manufacturer's instructions. *Procr* probes (REF#410321) and *Ccn1* probes (REF#429001) were ordered from Advanced Cell Diagnostics. After in situ hybridization, TSA method was used for Krt8 staining following the manufacturer's instructions using the Opal 4-Color Automation IHC Kit (PerkinElmer). The images were captured using Leica DM6000 TCS/SP8 laser confocal scanning microscope with a ×63/0.75 IMM objective.

## EdU labeling assays

The proliferation of OSE cells in vivo was measured by EdU uptake. Briefly, mice were injected with 100 μl EdU (2.5 mg/ml in dimethyl sulfoxide) for 12 hr. Then ovaries were harvested for section, following by EdU color staining using Click-iTEdU Alexa Fluor Imaging Kit (prepared according to the manufacturer's instructions). After washed with PBS for three times (10 min per time), EdU color development was performed following the manufacturer's protocol. After EdU signal developing, sections were blocked in blocking buffer for 1 hr at room temperature followed by antibody staining and mounted with mounting medium for imaging and quantification.

## Living image of cultured OSE cells

OSE cells were isolated from the mice and cultured on glass for 3–4 days. DOX was added into the medium 1 day and Hoechst 33342 was added 30 min before image. Live-cell imaging was performed at 37°C on a Zeiss Cell discoverer seven with perfect focus system. Cells were imaged at 1 time per 5 min for 24 hr with 70% laser power.

## Chromatin immunoprecipitation-qPCR

Cultured primary OSE cells were crosslinked in a final concentration of 1% formaldehyde (Sigma) PBS buffer for 15 min at 37°C, then added glycine to stop crosslinking. Chromatin from nuclei was sheared to 200–600 bp fragments using ultrasonic apparatus, then immunoprecipitated with antibody of TEAD4 (ab58310, Abcam) or normal mouse IgG (sc-2025, Santa Cruz) overnight. Antibody/antigen

complexes were recovered with Protein A/G PLUS-Agarose (sc-2003, Santa Cruz Biotechnology) for 2 hr at 4°C. After washing, the chromatin was eluted, decrosslinked and digested. The immuno-precipitated DNA was collected with QIAQIUCK PCR Purification Kit (QIAGEN). Purified DNA was performed with ChIP-qPCR. Assessing the enrichment of the proteins of interest on the targeting region by calculating the value of 'fold over IgG'.ChIP-qPCR primers used were as follows.

> Negative Ctrl CHIP-R, TATCCCCACTGCCCAGAAGA.
> Negative Ctrl CHIP-F, GATCAACGCAGGGGAGAGAG.
> Procr CHIP-R, GTGAATGCACACACACACCC.
> Procr CHIP-F, ATATCCGAGCTACACACGGC.
> Ctfg CHIP-R, GAACTGAATGGAGTCCTACACA.
> Ctfg CHIP-F, TGTGCCAGCTTTTTCAGACG.

## Preparation of Procr promoter luciferase reporter and luciferase assay

The DNA sequence of Procr promoter containing TEAD4-binding sites (about 2 kb before the initiation codon) were amplified by PCR, separated by agarose gel, purified by Gel Extraction Kit, and then cloned into pGL3-promoter vector. Luciferase assays were performed in 293T cells with the pGL3-Procr promoter luciferase reporter described above 0.2 mg reporter plasmid were transfected together with CMV-Renilla (0.005 mg) to normalize for transfection efficiency. For luciferase assays in overexpression plasmid-transfected cells, cells were transfected with the indicated plasmids and reporter plasmid together, and then the luciferase activity was measured 36 hr later using Dual-Luciferase Reporter Assay System Technical Manual kit following the manufacturer's protocol.

## Cell culture, viral production, and infection

HEK 293T was obtained from American Type Culture Collection (ATCC) and cultured in DMEM supplemented with 10% FBS plus 1% penicillin and streptomycin antibiotics at 37°C in 5% $CO_2$ (vol/vol). For cells cultured on different modulus of elasticity, hydrogel substrates with tunable mechanical properties were prepared following the previous protocol (*Tse and Engler, 2010*), and the glass was as solid control. HEK 293T cells were used to produce lentivirus. When cells were up to 80%–90%, indicated constructs and packaging plasmids transfection was performed in Opti-MEM, and the media were replaced 12 hr later. Viral supernatants were collected 48–72 hr after medium change and filtered through a 0.45-μm filter, followed by concentration. For primary OSE cells infection, concentrated virus was diluted in the culture medium along with 1:100 polybrene.

## RNA isolation and quantitative real-time PCR

Total RNA was isolated from fresh OSE cells or cultured cells lysed with Trizol according to the manufacturer's instructions. The cDNA was generated from equal amounts of RNA using the SuperScriptIII kit. qPCR was performed on a StepOne Plus (Applied Biosystems) with Power SYBR Green PCR Master Mix. RNA level was normalized to *Gapdh*. The cycling condition was as 10 min at 95°C for initial denaturing, 40 cycles of 15 s at 95°C for denaturing, 1 min at 60°C for annealing and extension, following by melt curve test.

## Quantification and statistical analysis

For quantification of nuclear Yap1+, Vgll4+, and EdU+ cells, 40 OSE cells at the both edges of ruptured sites (20 OSE cells at one side of rupture site) was identified as rupture regions, while other regions as nonrupture regions. At least 30 rupture regions and 30 nonrupture regions were counted. For quantification of the diameter of rupture, the longest diameter was counted, and at least 20 rupture sites were counted. For quantification of mG+ clone sizes, about 0.3 mm$^2$ circle centered on ruptured sites was identified as rupture regions. At least 30 rupture regions were counted. For quantification of colonies size, diameters of the colonies were measured using Zeiss software.

Statistical analyses were calculated in GraphPad Prism (Student's *t*-test or one-way analysis of variance). For all experiments with error bars, the standard error of measurement was calculated to indicate the variation within each experiment. All the p values were calculated using GraphPad PRISM six with the following significance: n.s. $p > 0.05$; *$p < 0.05$; **$p < 0.01$; ***$p < 0.001$. Statistical details for each experiment can be found in the figures and the legends.

## Acknowledgements

Thank to Dr. Chi Chung Hui of University of Toronto for helpful comments on the manuscript. This research was supported by grants from National Key Research and Development Program of China (2019YFA0802000 and 2020YFA0509002 to YAZ); National Natural Science Foundation of China (31625020 to YAZ; 32030025 and 31625017 to LZ); the Chinese Academy of Sciences (XDA16020200 to YAZ); Fundamental Research Funds for the Central Universities (2020XZZX002-22 to JF); the National Key Research and Development Programme of China (2021YFC2701901 to JF); China Postdoctoral Science Foundation (2020TQ0260 to JW); Zhejiang Provincial Preferential Postdoctoral Foundation (ZJ2020150 to JW); Postdoctoral Exchange Fellowship Program 2021 (No.160 Document of OCPC, 2021) to JW; Shanghai Leading Talents Program to LZ.

## Additional information

### Competing interests

Yi Arial Zeng: Reviewing editor, *eLife*. The other authors declare that no competing interests exist.

### Funding

| Funder | Grant reference number | Author |
| --- | --- | --- |
| National Key Research and Development Program of China | 2019YFA0802000 | Yi Arial Zeng |
| National Key Research and Development Program of China | 2020YFA0509002 | Yi Arial Zeng |
| National Natural Science Foundation of China | 31625020 | Yi Arial Zeng |
| Chinese Academy of Sciences | XDA16020200 | Yi Arial Zeng |
| National Natural Science Foundation of China | 32030025 | Lei Zhang |
| National Natural Science Foundation of China | 31625017 | Lei Zhang |
| Fundamental Research Funds for the Central Universities | 2020XZZX002-22 | Junfen Fu |
| National Key Research and Development Program of China | 2021YFC2701901 | Junfen Fu |
| China Postdoctoral Science Foundation | 2020TQ0260 | Jingqiang Wang |
| Zhejiang Provincial Preferential Postdoctoral Foundation | ZJ2020150 | Jingqiang Wang |
| Postdoctoral Exchange Fellowship Program 2021 | No. 160 Document of OCPC 2021 | Jingqiang Wang |

The funders had no role in study design, data collection, and interpretation, or the decision to submit the work for publication.

### Author contributions

Jingqiang Wang, Conceptualization, Data curation, Formal analysis, Funding acquisition, Investigation, Project administration, Validation, Writing - original draft, Writing – review and editing; Chunye Liu, Data curation, Project administration, Resources, Supervision, Validation; Lingli He, Zhiyao Xie, Wentao Yu, Data curation, Investigation; Lanyue Bai, Data curation, Resources; Zuoyun Wang, Yi Lu,

Resources; Dong Gao, Writing – review and editing; Junfen Fu, Funding acquisition, Supervision; Lei Zhang, Funding acquisition, Project administration, Resources, Supervision, Writing – review and editing; Yi Arial Zeng, Conceptualization, Formal analysis, Funding acquisition, Project administration, Resources, Supervision, Writing - original draft, Writing – review and editing

**Author ORCIDs**
Jingqiang Wang  http://orcid.org/0000-0003-1670-7759
Chunye Liu  http://orcid.org/0000-0001-7167-6735
Lei Zhang  http://orcid.org/0000-0003-2566-6493
Yi Arial Zeng  http://orcid.org/0000-0003-1898-8099

**Ethics**
All mice were housed in the SIBCB animal facility under IVC standard with a 12-hr light/dark cycle at room temperature. Experimental procedures were approved by the Animal Care and Use Committee of Shanghai Institute of Biochemistry and Cell Biology, Chinese Academy of Sciences, with a project license number of IBCB0065.

**Decision letter and Author response**
Decision letter https://doi.org/10.7554/eLife.75449.sa1
Author response https://doi.org/10.7554/eLife.75449.sa2

---

## Additional files

**Supplementary files**
• Transparent reporting form

**Data availability**
All data generated or analysed during this study are included in the manuscript and supporting file; Source Data files have been provided for Figure 1-source data 1, Figure 2-source data 1, Figure 3-source data 1, Figure 4-source data 1, Figure 5-source data 1, Figure 1-figure supplement 2-source data 1, Figure 2-figure supplement 1-source data 1,Figure 2-figure supplement 1-source data 2,Figure 4-figure supplement 1-source data 1;Figure 4-figure supplement 2-source data 1;Figure 4-figure supplement 2-source data 2.

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

# Appendix 1

## Appendix 1—key resources table

| Reagent type (species) or resource | Designation | Source or reference | Identifiers | Additional information |
|---|---|---|---|---|
| Genetic reagent (*M. musculus*) | Mouse: *Procr-CreER* | *Procr$^{CreER}$* | N/A | *Wang et al., 2015* |
| Genetic reagent (*M. musculus*) | Mouse: *Procr-rtTA* | *Procr$^{rtTA}$* | N/A | *Wang et al., 2019* |
| Genetic reagent (*M. musculus*) | Mouse: *R26-mTmG* | The Jackson Laboratory | Jax: 007576 | |
| Genetic reagent (*M. musculus*) | Mouse: *TetO-H2B-GFP* | The Jackson Laboratory | Jax: 005104 | |
| Genetic reagent (*M. musculus*) | Mouse: *Yap1$^{fl/+}$* | *Yap1$^{fl/+}$* | N/A | *Feng et al., 2019* |
| Genetic reagent (*M. musculus*) | Mouse: *Procr$^{fl/+}$* | Liu and Zeng, unpublished | N/A | Liu and Zeng, unpublished |
| Genetic reagent (*M. musculus*) | Mouse: ICR | SLAC | N/A | Female |
| Genetic reagent (*M. musculus*) | Mouse: C57BL/6 | SLAC | N/A | Female |
| Genetic reagent (*M. musculus*) | Mouse: *TetO-Vgll4-Flag* | This paper | N/A | See Materials and methods |
| Cell line (*Homo-sapiens*) | Human: HEK 293T cells | ATCC | Cat# CRL-3126 | |
| Antibody | anti-CD31, Biotin (Rat monoclonal) | BD | Cat# 553371; RRID:AB_394817 | FACS (1:200) |
| Antibody | anti-CD45, Biotin (Rat monoclonal) | BD | Cat# 553080; RRID:AB_394610 | FACS (1:200) |
| Antibody | anti-Ter119, Biotin (Rat monoclonal) | BD | Cat# 553672; RRID:AB_394985 | FACS (1:200) |
| Antibody | anti-CD31, FITC (Rat monoclonal) | BD | Cat# 553372; RRID:AB_394818 | FACS (1:200) |
| Antibody | anti-CD45, FITC (Rat monoclonal) | BD | Cat# 553079; RRID:AB_394609 | FACS (1:200) |
| Antibody | anti-Ter119, FITC (Rat monoclonal) | BD | Cat# 557915; RRID:AB_396936 | FACS (1:200) |
| Antibody | anti-EpCAM, APC (Rat monoclonal) | Thermo Fisher | Cat# 17-5791-82; RRID:AB_2716944 | FACS (1:200) |
| Antibody | Streptavidin-apc-Cy7 | BioLegend | Cat# 405208; RRID:N/A | FACS (1:500) |
| Antibody | Streptavidin-V450 | BD | Cat# 560797; RRID:AB_2033992 | FACS (1:500) |
| Antibody | anti Procr, Biotin (Rat monoclonal) | Thermo Fisher | Cat# 13-2012-82; RRID:AB_657694 | FACS (1:200) |
| Antibody | anti Procr, PE (Rat monoclonal) | Thermo Fisher | Cat# 12-2012-82; RRID:AB_914317 | FACS (1:200) |
| Antibody | anti-Krt8 (Rat monoclonal) | DSHB | Cat# TROMA-I; RRID:AB_531826 | IHC (1:250) |
| Antibody | anti-E-Cadherin (Mouse monoclonal) | BD | Cat# 610181; RRID:AB_397581 | IHC (1:100) |
| Antibody | anti-Yap1 (Rabbit monoclonal) | CST | Cat# 14074; RRID:AB_2650491 | IHC (1:100) |
| Antibody | anti-Yap1 (Rabbitpolyclonal) | ABclonal | Cat# A1002; RRID:AB_2757539 | IHC (1:200) WB (1:200) |
| Antibody | anti-GFP (Chicken polyclonal) | Thermo fisher | Cat# A10262; RRID:AB_2534023 | IHC (1:500) |
| Antibody | anti-Vgll4 (Rabbit polyclonal) | Self-made | Cat# N/A; RRID:N/A | IHC (1:100) |

*Appendix 1 Continued on next page*

*Appendix 1 Continued*

| Reagent type (species) or resource | Designation | Source or reference | Identifiers | Additional information |
|---|---|---|---|---|
| Antibody | anti-Flag (Mouse monoclonal) | Sigma | Cat# F1804; RRID:AB_262044 | WB (1:100) |
| Antibody | anti-Vgll4 (Rabbitpolyclonal) | ABclonal | Cat# A18248; RRID:AB_2862024 | IHC (1:100) WB (1:100) |
| Antibody | anti-Laminin (Rabbit polyclonal) | Sigma | Cat# L9393; RRID:AB_477163 | IHC (1:500) |
| Antibody | anti-a-E-catenin (Rabbit polyclonal) | Proteintech | Cat# 12831-1-AP; RRID:AB_2087822 | IHC (1:200) |
| Antibody | anti-ZO-1 (Mouse monoclonal) | Thermo Fisher | Cat# 33-9100; RRID:AB_2533147 | IHC (1:100) |
| Antibody | Normal mouse IgG | Santa cruz | Cat# sc-2025; RRID:AB_737182 | ChIP (1:100) |
| Antibody | anti-TEAD4 (Mouse monoclonal) | Abcam | Cat# ab58310; RRID:AB_945789 | ChIP (1:100) |
| Recombinant DNA reagent | pLKO.1-EGFP-shYap1 | This paper | Plasmid | Pol III-based shRNA backbone |
| Recombinant DNA reagent | pcDNA3.1-Yap1 | This paper | Plasmid | pcDNA3.1 backbone |
| Recombinant DNA reagent | PGL3.1 Procr promoter | This paper | Plasmid | PGL3.1 Promoter backbone |
| Recombinant DNA reagent | PGL3.1 Procr promoter TEAD4-binding motif mutation | This paper | Plasmid | Deletion of 5′CATTCC3′ at the −1486 to −1481 bp in Procr Promoter region |
| Sequence-based reagent | Procr-F | This paper | qPCR primers | 5′CTCTCTGGGA AAACTCCTGACA3′ |
| Sequence-based reagent | Procr-R | This paper | qPCR primers | 5′CAGGGAGCAGCT AACAGTGA3′ |
| Sequence-based reagent | Vgll4-F | This paper | qPCR primers | 5′ATGAACAACAATA TCGGCGTTCT3′ |
| Sequence-based reagent | Vgll4-R | This paper | qPCR primers | 5′GGGCTCCATGCT GAATTTCC3′ |
| Sequence-based reagent | Yap1-F | This paper | qPCR primers | 5′GCCATGCTTTCG CAACTGAA3′ |
| Sequence-based reagent | Yap1-R | This paper | qPCR primers | 5′CAAAACGAGGGT CCAGCCTT3′ |
| Sequence-based reagent | Ccn1-F | This paper | qPCR primers | 5′TCGCAATTGGAA AAGGCAGC3′ |
| Sequence-based reagent | Ccn1-R | This paper | qPCR primers | 5′CCAAGACGTGG TCTGAACGA3′ |
| Sequence-based reagent | Birc5-F | This paper | qPCR primers | 5′AGAACAAAATTG CAAAGGAGACCA3′ |
| Sequence-based reagent | Birc5-R | This paper | qPCR primers | 5′GGCATGTCAC TCAGGTCCAA3′ |
| Commercial assay or kit | Click-iTTMEdU Cell Proliferation Kit for Imaging | Thermo Fisher | Cat# C10337 | |
| Commercial assay or kit | Dual-Luciferase Reporter Assay System Technical Manual | Promega | Cat# E1910 | |
| Commercial assay or kit | Opal 4-Color Automation IHC Kit | PerkinElmer | Cat# NEL8200001KT | |
| Commercial assay or kit | RNAscope Multiplex Fluorescent Detection Kit v2 | ACD | Cat# 323,110 | |
| Commercial assay or kit | SYBR green Mix | Roche | Cat# 04913914001 | |
| Commercial assay or kit | PrimeScript RT master Mix | Takara | Cat# RR036A | |
| Chemical compound, drug | Hoechst33342 | Thermo Fisher | Cat# H21492 | |
| Chemical compound, drug | Tamoxifen | Sigma-Aldrich | Cat# T5648 | |

*Appendix 1 Continued on next page*

*Appendix 1 Continued*

| Reagent type (species) or resource | Designation | Source or reference | Identifiers | Additional information |
|---|---|---|---|---|
| Chemical compound, drug | Doxcyclinehyclate | Sigma-Aldrich | Cat# D9891 | |
| Chemical compound, drug | Verteporfin | MCE | Cat# HY-B0146 | |
| Software, algorithm | Flow Jo vX | Flow Jo | https://www.flowjo.com | |
| Software, algorithm | GraphPad Prism 6 | GraphPad software | https://www.graphpad.com | |
| Other | DNase I | Sigma-Aldrich | Cat# D4263 | |
| Other | Type IV collagenase | Worthington | Cat# LS004189 | |
| Other | Matrigel | BD | Cat# 354,230 | |
| Other | Dispase | BD | Cat# 354,235 | |
| Other | DAPI | Thermo Fisher | Cat# D1306 | |
| Other | MEM Non-Essential Amino Acids Solution | Thermo Fisher | Cat# 11140050 | |
| Other | L-Glutamine | Thermo Fisher | Cat# 25030081 | |
| Other | ITS | Thermo Fisher | Cat# 41,440 | |
| Other | 2-Mercaptoethanol | Millipore | Cat# ES-007-E | |
| Other | Hydrocortisone | Sigma-Aldrich | Cat# 614,517 | |
| Other | Sodium pyruvate | Thermo Fisher | Cat# 11360070 | |

