## [Editor Report]

This is a well conducted and interesting study that increases our knowledge of mechanisms governing healing after ovarian rupture. This study specifically demonstrates the importance of Procr+ progenitors that are positively regulated by YAP signalling.

---

## [Decision Letter]

**Decision letter after peer review:**

Thank you for submitting your article "Selective YAP activation in Procr cells is essential for ovarian stem/progenitor expansion and epithelium repair" for consideration by *eLife*. Your article has been reviewed by 3 peer reviewers, one of whom is a member of our Board of Reviewing Editors, and the evaluation has been overseen by Ricardo Azziz as the Senior Editor. The reviewers have opted to remain anonymous.

Essential revisions:

The reviewers were generally positive about this manuscript. The requested revisions are mostly editorial and involve tempering claims and/or editing the text or figures. Simple experiments, designed to strengthen conclusions are also requested.

1) The authors failed to identify how stem/progenitor cells sense ovarian rupture to then trigger rupture repair through inducing pathways such as Yap. It is not necessary that the authors identify what senses the rupture for publication, as identifying the sensors that detect tissue wounding is a major challenge in regenerative biology. However, the authors claimed to have resolved this point in the abstract, introduction and discussion, and therefore should be more precise in describing their main findings.

2) If the ovulation hormones used to induce ovarian tissue rupture can potentially induce Yap signaling, the authors should explicitly discuss this potential experimental limitation in their manuscript.

3) The authors conclude that YAP signalling mediates its effects on healing at least in part by enhancing the expression of Procr in OSE. However, as it stands the results remain correlative. The authors should thus determine whether Procr overexpression can overcome YAP KO or Vgll4 overexpression, at least using in vitro models.

4) YAP pathway activation was extrapolated using in situ hybridization for one target (Cyr61). If feasible, the authors are encouraged to examine additional YAP signalling target genes by the same or another independent method (such as qPCR), to strengthen their main conclusion.

5) In Line 115, the authors state "various adherent or tight junction components" were examined. However, the results showed the effect on E-cadherin only. Did the authors examine multiple components or just one? Additional adherens junction proteins besides E-cadherin (such as a-catenin) would be helpful to reinforce their conclusion.

6) Please further characterise or describe the sorted OSE cells (perhaps with additional markers) to ensure that the Procr+ and Procr- cells are of the same lineage.

7) For figure 4 l and m, please show IF and/or Westerns to validated increasing YAP expression.

8) Resolution of the images on YAP immunostaining (Figure 1d, 1e) was too poor to discriminate between nuclear or cytoplasmic staining patterns. Please improve these images.

9) Please quantify proliferation shown in Figure S3b.

10) Line 303, "bigger openings compared to control" the subject was reversed in Figure legends "would size in Ctrl were smaller". Please correct

11) Line 303, 7d pi did not match Figure 5e, which was labeled as 7.5d. Please correct.

12) Results in lines 307 and 308 (5.5, 1,4) did not match legends of Figure 5h (4.73, 1.62. Please correct.

*Reviewer #1:*

This is a well conducted and interesting study that increases our knowledge of mechanisms governing healing after ovarian rupture. This study specifically demonstrates the importance of Procr+ progenitors that are positively regulated by YAP signaling. The study uses elegant mouse models and for the results are convincing. A few revisions would enhance the study and enable the authors to more strongly conclude that YAP promotes healing at least in part by increasing the expression of Procr.

Col1a1 is not specific for Procr+ OSE cells. I do not think this will affect the results terribly, but should be acknowledged.

The authors conclude that YAP signalling mediates its effects on healing at least in part by enhancing the expression of Procr in OSE. However, as it stands the results remain correlative. The authors should thus determine whether Procr overexpression can overcome YAP KO or Vgll4 overexpression, at least using in vitro models.

The authors should attempt to determine whether there are human correlates to the phenomena observed in the mouse models.

Please further characterise the sorted OSE cells (perhaps with additional markers) to ensure that the Procr+ and Procr- cells are pf the same lineage.

For figure 4 l and m, please show IF and/or Westerns to validated increasing YAP expression.

Please quantify proliferation shown in Figure S3b.

*Reviewer #2:*

The ovary is a fascinating yet understudied organ in the field of stem cell biology: every reproductive cycle, it is ruptured as eggs are released. The ovary thus undergoes successive waves of injury. The mechanisms through which the ovary's surface epithelium is repaired remain incompletely understood. Multiple candidate stem and progenitor cells have been described for the ovarian surface epithelium (Ng and Barker, 2015; Nat Rev Mol Cell Biol), and the authors of the present report have previously demonstrated that Procr+ progenitors contribute to ovarian rupture repair (Wang et al., 2019; Nature Communications).

In the present report, the authors attempt to answer the fascinating question of how progenitors sense tissue injury in order to effect regeneration; broadly speaking, how tissues sense injury is one of the biggest questions in the field of stem cell biology. The authors show that the transcriptional coactivator Yap is expressed by Procr+ ovarian epithelial progenitors, and is necessary for rupture repair. This is demonstrated through two parallel methods: directly deleting Yap (Figure 2) or overexpressing Vgll4, which encodes a Yap antagonist (Figure 3). Both of these genetic manipulations are specifically conducted in Procr+ ovarian epithelial progenitors in vivo.

While the authors convincingly show that Yap is necessary for ovarian rupture repair, the biggest unanswered question in the paper is, mechanistically, how the rupture is actually sensed (i.e., what turns on Yap?) Yap is a transcriptional coactivator that is repressed by the Hippo mechanotransduction pathway, and there are multiple mechanical inputs (e.g., cell adhesion, cell stretching, extracellular matrix stiffness, and cell density) that can all regulate the Hippo pathway (Halder et al., 2012; Nat Rev Mol Cell Biol). The authors claim that the E-cadherin cell-adhesion protein is lowered at the site of the ovarian rupture (Figure 1), and it was previously known that E-cadherin signals through the Hippo pathway to block Yap (Yang et al., 2015; PNAS).

However the missing link in the paper is why E-cadherin would be reduced at the ovarian rupture site. There are other potentially more direct possibilities. For instance, it is known that mechanical stretching induces Yap (Halder et al., 2012; Nat Rev Mol Cell Biol). First, during the process of egg release, overlying epithelial cells will possibly become "stretched", which would induce Yap? Second, at the rupture site, epithelial cells will no longer become packed together because the epithelium has been denuded: so potentially cells flanking the rupture site would become "stretched", which would also activate Yap? These ideas are speculative, but are intended to try to assist the authors in formulating a mechanistic link between ovarian ruptures and Yap activation. For instance, the authors could consider looking at cell stretching/roundness/morphology by imaging.

In summary, the authors show that Yap is required for ovarian regeneration, consistent with the well-known roles of Yap in cell proliferation and tissue repair in other contexts (Ramos and Camargo, 2012; Trends in Cell Biology). However the authors may not have fully addressed the question they posed in the Introduction, namely how is ovarian rupture sensed and how is Yap itself induced by injury.

Comments for the authors:

There are multiple merits to this paper, including the demonstration that Yap is required in Procr+ ovarian progenitors to drive rupture repair. However the missing link in the manuscript is how tissue rupture is sensed and how Yap is turned on: something that the authors raised in the Introduction of their paper. The authors show E-cadherin is decreased at the rupture (Figure 1), but it is unclear why E-cadherin should be decreased.

An open question remains whether the authors need to find the "rupture sensing mechanism" (i.e., what is upstream of Yap) to warrant publication in *eLife*. It is also possible that without finding this mechanism that the manuscript could be published. But in that case, the authors may want to tone down their claims that they discovered how "stem cell senses the rupture and promptly turns on proliferation".

Strengths:

1. The ovary is an understudied organ, and this study sheds light onto its regenerative powers.

2. The authors specifically examine the role of Yap in a defined cell population (Procr+ progenitors), instead of perturbing Yap in the whole organ.

3. Two parallel approaches are used to block Yap: directly deleting Yap (Figure 2) or overexpressing Vgll4, which encodes a Yap antagonist (Figure 3). Both of these perturbations compromise effective rupture repair. Yap deletion also reduces the frequency of Procr+ progenitors (Figure 4).

Weaknesses:

1. Why is E-cadherin decreased at rupture sites? (Figure 1a)

2. The authors suggest that nuclear Yap is induced in superficial epithelial cells nearby the rupture (Figure 1b). While the higher-magnification images are a bit easier to interpret, the zoomed-out image (leftmost image in Figure 1b) is not that clear. Yap seems to be expressed in many cells, including deeper cells (i.e., not superficial epithelial cells) and even some cells quite far away from the rupture.

3. The authors claim that reduced Vgll4 levels in Procr+ progenitors (as opposed to Procr- non-progenitors) renders Procr+ progenitors hyper-sensitive to Yap activation. However there is a fairly modest ~2-fold difference in Vgll4 mRNA between Procr+ vs. Procr- cells (Figure 3b) – is this enough to substantially change a cell's receptiveness to Yap signaling?

4. The notion that Procr itself is required for ovarian regeneration is potentially interesting (Figure 5). However, Procr is expressed in multiple cell-types within the ovary; the authors actually showed that vascular cells are a major population of Procr+ cells (Wang et al., 2019; Nature Communications). Is it possible that when Procr is deleted that the epithelial progenitors themselves are not significantly impacted, but rather the impaired regeneration can be ascribed to vascular (or other) defects? Procr-/- mice die of vascular defects (Gu et e al., 2002; J Biol Chem).

*Reviewer #3:*

The wound healing process during the normal ovulation process is a very interesting but poorly understood process. Wang et al. intend to identify molecular mechanisms by which tissue-resident stem/progenitor cells engage in response to epithelium rupture. Wang et al. made use of a marker gene, Procr, to investigate stem/progenitor population in the ovarian epithelium that was described in their earlier publication. With the advantage of being able to isolate and characterize this population, strengths of the methods employed in the manuscript involves the usage of several novel mouse strains to perform lineage tracing or genetic ablation specifically in the Procr+ cells versus other epithelial cells that lack Procr expression. The strengths of the results in this study revealed compelling functional phenotypes in respect to loss of function (YAP, Procr) and gain of function (Vgll4) in the context of ovarian rupture repair process.

The weakness of the methods is the means of inducing ovarian epithelium rupture requires the usage of two hormones, which can also activate YAP pathway independently. Although this does not have a large impact on the conclusions, it is important to recognize the observation on YAP pathway is associated with this specific experimental method. However, multiple methods were used to identify functional consequences of ovarian rupture repair; the conclusions are well supported by the results. In addition, the authors generated several new transgenic mouse strains (Procr-rtTA, TetO-Vgll4, Procrflox) to address the functional relevance of cell type of origin as well as the roles of YAP signaling pathway. These strains will be useful resources for the community.

1. The title and main conclusion of the study on selective activation of the YAP pathway within Procr+ progenitors were supported by the detection of a single target Cyr61 examined by RNA in situ hybridization. If feasible, the authors are encouraged to examine additional target genes by the same or another independent method (such as qPCR), to strengthen their main conclusion.

2. In Line 115, the authors state "various adherent or tight junction components" were examined. However, the results showed the effect on E-cadherin only. Did the authors examine multiple components or just one? Additional adherens junction proteins besides E-cadherin (such as a-catenin) would be helpful to reinforce their conclusion.

3. Resolution of the images on YAP immunostaining (Figure 1d, 1e) was too poor to discriminate between nuclear or cytoplasmic staining patterns.

---

## [Author Response]

Essential revisions:The reviewers were generally positive about this manuscript. The requested revisions are mostly editorial and involve tempering claims and/or editing the text or figures. Simple experiments, designed to strengthen conclusions are also requested.1) The authors failed to identify how stem/progenitor cells sense ovarian rupture to then trigger rupture repair through inducing pathways such as Yap. It is not necessary that the authors identify what senses the rupture for publication, as identifying the sensors that detect tissue wounding is a major challenge in regenerative biology. However, the authors claimed to have resolved this point in the abstract, introduction and discussion, and therefore should be more precise in describing their main findings.

We apologize for the confusion caused. Following the suggestion, we have rephrased “senses” into “is triggered by” in the abstract, into “the molecular mechanism in which linking the ovulatory rupture to OSE stem/progenitor cells activation” in the introduction, and discussion. We especially appreciate the reviewers’ insights on the possible reasons for turning on Yap, and we have included them in the discussion (p.13-14).

“While we uncovered the significance of selective activation of Yap in OSE progenitors, it is still unclear how ovarian rupture is sensed and how Yap signaling is induced by injury. […] Second, at the rupture site, epithelial cells no longer become packed together because the epithelium has been denuded, therefore potentially cells flanking the rupture site would become "stretched", consequently actives YAP.”

2) If the ovulation hormones used to induce ovarian tissue rupture can potentially induce Yap signaling, the authors should explicitly discuss this potential experimental limitation in their manuscript.

Following the advice, we found two pieces of literature. One reports that HCG down-regulates YAP1 protein level and target gene *Ctgf* expression in granulosa cells ^1^; another one shows that HCG downregulates YAP1 expression and upregulates p-YAP in cumulus-oocyte complexes ^2^. Thus, both papers suggest ovulation hormones suppress Yap signaling in granulosa cells. We have not come across previous reports about Yap and OSE.

3) The authors conclude that YAP signalling mediates its effects on healing at least in part by enhancing the expression of Procr in OSE. However, as it stands the results remain correlative. The authors should thus determine whether Procr overexpression can overcome YAP KO or Vgll4 overexpression, at least using in vitro models.

We appreciate the reviewer’s suggestion. We would like to clarify that in the proposed model, YAP signaling activation at the rupture site drives cell division and simultaneously enhances Procr expression (Figure 5m). Therefore, we reasoned that overexpression of Procr alone would not be sufficient to rescue YAP-cKO or Vgll4-OE.

Nevertheless, we understand the reviewers’ concern and performed relevant experiments, in which we overexpress YAP and knockdown Procr in a human OSE cell line, OVCAR3 cells. Overexpression of YAP promoted the in vitro sphere formation and in vivo xenografts growth. While knockdown of PROCR blocked the sphere formation and xenografts growth induced by YAP-OE. Those data suggest YAP signaling mediates its effects in part by PROCR (see Author response image 1).

**Author response image 1. sa2fig1:** 

4) YAP pathway activation was extrapolated using in situ hybridization for one target (Cyr61). If feasible, the authors are encouraged to examine additional YAP signalling target genes by the same or another independent method (such as qPCR), to strengthen their main conclusion.

We appreciate the suggestion. We included qPCR analyses in the revised Figure S2c, showing that Procr+ OSE cells have a higher level of *Birc5* expression, another YAP target gene.

5) In Line 115, the authors state "various adherent or tight junction components" were examined. However, the results showed the effect on E-cadherin only. Did the authors examine multiple components or just one? Additional adherens junction proteins besides E-cadherin (such as a-catenin) would be helpful to reinforce their conclusion.

Following the suggestion, we included the immunostaining of adherents and tight junction proteins in the revised Figure S1c-d, showing a decreased α-catenin and ZO-1 in the OSE of rupture proximal regions compared to rupture distal and non-rupture regions.

6) Please further characterise or describe the sorted OSE cells (perhaps with additional markers) to ensure that the Procr+ and Procr- cells are of the same lineage.

Following the suggestion, we performed Krt8 (K8) staining with the sorted Procr+ and Procr- OSE cells, and the results indicated that both Procr+ and Procr- OSE are K8+ epithelial cells. These results have been included in revised Figure S2a-b.

7) For figure 4 l and m, please show IF and/or Westerns to validated increasing YAP expression.

Following the suggestion, we performed immunostaining and Western blotting, and the results verified increasing YAP expression. These data have been included in the revised Figure S5h-i.

8) Resolution of the images on YAP immunostaining (Figure 1d, 1e) was too poor to discriminate between nuclear or cytoplasmic staining patterns. Please improve these images.

Following the suggestion, we have replaced them with images with higher magnification in revised Figure1d-e.

9) Please quantify proliferation shown in Figure S3b.

Following the suggestion, we have quantified the proliferation of Figure S4a-b (previous S3a-b) and shown in the revised Figure S4c.

10) Line 303, "bigger openings compared to control" the subject was reversed in Figure legends "would size in Ctrl were smaller". Please correct

We apologize for the confusion caused. It has been revised to “control ovaries showed smaller openings compared to Procr-cKO (Figure 5c, 5e)” in Line 341 (previous Line 303).

11) Line 303, 7d pi did not match Figure 5e, which was labeled as 7.5d. Please correct.

We apologize for the mistake. It has been revised to 7.5d.

12) Results in lines 307 and 308 (5.5, 1,4) did not match legends of Figure 5h (4.73, 1.62. Please correct.

We apologize for the mistake. It has been revised to (4.7±0.4, 1.6±0.2).

Reference:

1. Ji, S.-Y. et al. The polycystic ovary syndrome-associated gene Yap1 is regulated by gonadotropins and sex steroid hormones in hyperandrogenism-induced oligo-ovulation in mouse. MHR: Basic science of reproductive medicine 23, 698-707, doi:10.1093/molehr/gax046 (2017).

2. Sun, T. and Diaz, F. J. Ovulatory signals alter granulosa cell behavior through YAP1 signaling. Reproductive Biology and Endocrinology 17, doi:10.1186/s12958-019-0552-1 (2019).